# Fragmented micro-growth habitats present opportunities for alternative competitive outcomes

Maxime Batsch [1], Isaline Guex[2], Helena Todorov[1], Clara M. Heiman [1], Jordan Vacheron [1], Julia A. Vorholt[3], Christoph Keel [1] & Jan Roelof van der Meer [1]✉

Bacteria in nature often thrive in fragmented environments, like soil pores, plant roots or plant leaves, leading to smaller isolated habitats, shared with fewer species. This spatial fragmentation can significantly influence bacterial interactions, affecting overall community diversity. To investigate this, we contrast paired bacterial growth in tiny picoliter droplets (1–3 cells per 35 pL up to 3–8 cells per species in 268 pL) with larger, uniform liquid cultures (about 2 million cells per 140 μl). We test four interaction scenarios using different bacterial strains: substrate competition, substrate independence, growth inhibition, and cell killing. In fragmented environments, interaction outcomes are more variable and sometimes even reverse compared to larger uniform cultures. Both experiments and simulations show that these differences stem mostly from variation in initial cell population growth phenotypes and their sizes. These effects are most significant with the smallest starting cell populations and lessen as population size increases. Simulations suggest that slower-growing species might survive competition by increasing growth variability. Our findings reveal how microhabitat fragmentation promotes diverse bacterial interaction outcomes, contributing to greater species diversity under competitive conditions.

The implications of habitat fragmentation on biodiversity constitute a widely investigated yet highly debated topic in classical macroecology[1]. While some authors have associated fragmentation with decreasing biodiversity in general[2], others have reported positive effects on biodiversity at the scale of landscapes[3,4]. In contrast, the aspect of habitat fragmentation has received only modest attention in microbial ecology[5,6]. The formation and maintenance of taxonomically rich microbial communities in nature (e.g. bacteria[7], archaea[8], microbial eukaryotes[9,10]) are thought to depend on the physicochemical conditions prevailing in their habitat[11,12], on inherent growth characteristics of the inhabiting species[13], and dynamic interspecific interactions that

emerge from shared nutrient and spatial niches[14,15]. However, most of our understanding of interspecific interactions comes from experimental studies using macro-habitats, controlled uniform growth conditions and large population census (>10⁶ cells per mL)[13,16–18], which are not necessarily representative for typically highly heterogenous natural microbial habitats (e.g. soil, plant leaves, skin). Thus, studies that take habitat heterogeneities into account are needed to extrapolate the roles of interspecific interaction effects on the development and maintenance of natural microbial communities.

Habitat heterogeneities at the scales relevant to microbial life occur in the form of spatial discontinuities and fragmentation[19–21],

[1]Department of Fundamental Microbiology, University of Lausanne, CH-1015 Lausanne, Switzerland. [2]Department of Mathematics, University of Fribourg, CH-1700 Fribourg, Switzerland. [3]Institute for Microbiology, Swiss Federal Institute of Technology (ETH Zürich), CH-8049 Zürich, Switzerland. ✉e-mail: Janroelof.vandermeer@unil.ch

which may have important implications for microbial community assembly and diversity[5,6,22–25]. For example, Conwill et al.[26], showed how different *Cutibacterium acnes* strains coexist at the macroscopic level of the human skin microbiome, but each individually colonizes a single skin pore; the pores creating niches without direct space and nutrient competition. Similarly, the particle structure of the soil habitat also creates multitudes of micro-pores and channels[22,27], which, dependent on water content, generate fluctuating networks of physically restricted and connected micro-growth environments[20,28]. Microhabitats also arise in animal guts, because of the physical shape of the epithelial cell lining (e.g., crypts[29]), the peristaltic motion of differently-sized food particles[30] or encapsulation of bacteria within gut mucus[31]. Surfaces of plant leaves have pronounced microstructures and properties leading to the formation of disconnected micro-droplets and water-filled channels, depending on humidity conditions[19,32,33]. Finally, even aquatic environments, considered to be connected, are characterized by plentiful particulate organic matter to which microbes attach, forming segregated habitats[34,35].

Habitat space constrains the opportunities for cells to get into physical proximity to every other member of the community. Therefore, even though the diversity in a macro-environment (e.g., soil) may be high (up to thousands of taxa[7]), the spatial discontinuities lead to microhabitat fragmentation, each containing perhaps a few or a few dozens of cells from only a limited number of taxa[27]. Communities in that regard are rather ensembles of myriads of smaller sub-communities inhabiting microhabitats. As a consequence, the assumed global roles of high-complexity interspecific interactions for community development would in individual microhabitats forcibly reduce to a few co-existing species with a smaller repertoire of potential interspecific interaction outcomes. In addition, because of the lower population census in microhabitats, one would expect phenotypic heterogeneity to play a more important role than in well-mixed conditions and at high population densities, where it would level out differences among individual founder cells. Indeed, measurements of individual phenotypic variations at the single (bacterial) cell level[36] show heterogeneous behaviour within low-census bacterial populations[37–39], and important heterogeneity in single cell growth kinetics[40–42]. Our hypothesis here was thus that microhabitats would enable existing phenotypic variation among individual founder cells (i.e., the cells present in a habitat that give rise to new offspring), to propagate into different reproductive success. In case of founder cells being part of a low-census multispecies community, we would then expect that growth kinetic variation could also lead to alternative outcomes of individual species growth and community composition. Expected interspecific interaction types from macro-scale experiments would then insufficiently explain their influence under microhabitat fragmentation. If true, growth variations in microhabitats could thus form an important driving force to sustain high observed microbial diversity despite competition for substrates being expected to drive poorly competitive species to extinction[13,43].

The main objective of this work was to study the effects of habitat fragmentation on bacterial community growth considering existing phenotypic heterogeneity. We tested our conjectures in four scenarios of different (expected) interspecific interaction types: (i) direct substrate competition, (ii) substrate independence, (iii) antagonism by inhibitory compounds and (iv) direct cell killing (Fig. 1a). Strain pairs were cultured either alone or in combination, and either in standard mixed liquid suspended culture (uniform growth conditions) with a large starting population census to observe global interaction types, or in fragmented microhabitats with each between 1–3 founder cells (Fig. 1b). Parallel fragmented microhabitats were created by emulsifying cell suspensions in growth medium into water-in-oil picoliter-droplets (35 or 268 pL) using a microfluidic device (Supplementary Fig. 1, design from Duarte et al.[44]). Water-in-oil droplets have been shown previously to restrict cell movement and diffusion of compounds between droplets, and effectively shield individual growth environments[45]. The cells in millions of generated droplets were subsequently incubated as an emulsion, and taxa growth was compared between mixed liquid suspension and pL-droplets. *Pseudomonas putida*[46] and *Pseudomonas veronii*[47] were used to test direct substrate competition and substrate independence. *Pseudomonas* Leaf 15, a known phyllosphere isolate producing inhibitory compounds[48] was tested with *Sphingomonas wittichi* RW1[49], for antagonistic interaction effects of diffusible compounds. Finally, *Pseudomonas protegens* strains CHA0 and Pf-5[50] were used to test the effect of tailocin-mediated killing. All strains were fluorescently tagged to be able to measure their productivity in individual droplets from microscopy imaging (Fig. 1c, d). Mathematical models describing competitive Monod growth[51] were used to examine the effects of stochastic founder cell and kinetic parameter variations on competitive strain dominance. Finally, we simulated and experimentally tested how the variation of interaction outcomes is influenced by the starting cell numbers within the microhabitats. Our results indicate that microhabitat fragmentation offers ecological opportunities to reverse interaction outcomes when the number of founder cells of each species is relatively low (smaller than ca. 10), leading to increased survival of poorly competitive strains in fragmented than in well-mixed uniform bulk environments.

## Results

### Growth in fragmented habitats enables local reversion of substrate competition

To understand how habitat micro-fragmentation affects the developing interactions between paired bacterial strains, we compared growth of mono- and cocultures under different interspecific interaction scenarios, and either in regular mixed liquid suspension with uniform growth conditions (with a large founder population size of $5 \times 10^5$ cells per 140 μL) or in pL droplets (with 1–3 founder cells per droplet and strain; Fig. 1a, b, Supplementary Fig. 2). In the first scenario, we focused on either substrate competition (i.e., a single shared primary carbon growth substrate in the form of succinate) or substrate independence (i.e., each strain receives its own specific substrate). To test this, we used two *Pseudomonas* strains (*P. putida* and *P. veronii*) with overlapping substrate preferences but different growth kinetic properties[51].

Growth rates of *P. putida* in uniform liquid-suspended culture with 10 mM succinate based on fluorescence measurements ($n = 6$ replicates) were slightly but significantly ($p = 0.009$) higher in mono than in co-culture with *P. veronii* (Fig. 2a, b; biomass yields from culture turbidity presented in Supplementary Fig. 3). In contrast, *P. veronii* grew slightly faster in co-culture with *P. putida* than alone ($p = 3.06 \times 10^{-4}$), possibly because of metabolite cross-feeding as suggested previously[51]. Despite this increase in co-culture, the average maximum specific growth rate ($\mu_{max}$) of *P. veronii* on succinate was 25% lower than that of *P. putida*, and the onset of growth (population lag time) was prolonged (Fig. 2a, b, Supplementary Fig. 3). Consequently, uniform liquid-suspended co-cultures became dominated by *P. putida* (Fig. 2a and c). Consistent with substrate competition, the strain-specific cell yields were lower in co- than in mono-cultures (Fig. 2c), with *P. putida* losing ca. 14.4% of its cell yield and *P. veronii* losing 84.7% compared to mono-cultures (Fig. 2c). Growth of the same cell suspension densities and substrate concentrations under fragmented conditions in pL-droplets both in mono- and co-cultures yielded similar cell numbers for *P. putida* in comparison to the suspended cultures (Fig. 2d, cell numbers determined after 24 h growth by flow cytometry by coalescing droplet emulsions into a single suspension). In contrast, the *P. veronii* cell numbers in droplet co-cultures with *P. putida* were on average 4 times higher than expected from suspended cultures (Fig. 2d). This suggested that the global competitive deficit of *P. veronii* in uniform liquid suspended culture was partly abolished during growth in fragmented conditions.

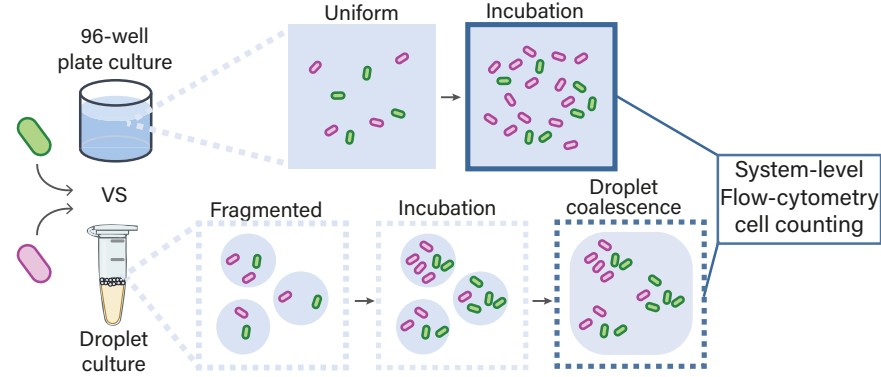

## a GROWTH INTERACTION SCENARIOS

Substrate competition — Substrate independence — Antibiotic-based competition — Tailocin-based competition

## b UNIFORM SUSPENDED VS DROPLET (FRAGMENTED) GROWTH

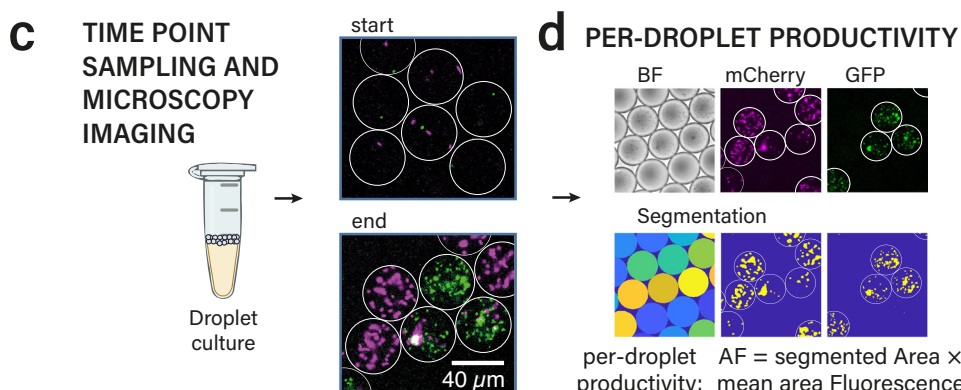

## c TIME POINT SAMPLING AND MICROSCOPY IMAGING

Droplet culture

start

end

40 μm

## d PER-DROPLET PRODUCTIVITY

BF    mCherry    GFP

Segmentation

per-droplet productivity: AF = segmented Area × mean area Fluorescence

**Fig. 1 | Experimental design. a** Four different pairwise interaction scenarios involving three different pairs of fluorescently tagged bacterial strains tested for growth and competition outcomes. **b** Uniform culturing in liquid suspensions in 96-well plates (ca. 2 million cells per strain at start in 140 μl) contrasted to that in fragmented droplets (0–3 cells per strain per droplet of 35 pL). Cell counts of both strain populations in stationary phase compared by flow cytometry. **c** Droplet emulsion sampling and microscopy imaging of individual droplets. **d** Productivity measurements of each strain by image segmentation of their specific fluorescent label within individual segmented droplets. Productivity (proxy for strain biomass) is reported as the product of the segmented strain-specific fluorescent area (in pixels, per droplet) times its droplet-mean fluorescent signal (called AF, for Area × Fluorescence).

To better understand the mechanisms for the attenuated competitive inhibition of *P. veronii* by *P. putida* in co-culture droplets, we looked more closely at the cell yield variations at the level of individual droplets (Fig. 2e). The median productivities of *P. putida* after 24 h growth in pL-droplets were indifferent between mono-cultures and droplets with only *P. putida* in co-cultures (Fig. 2e, *solo* droplets, $p = 0.2164$, $n = 3$, two-sided t-test), indicating that there was no difference arising from the co-culturing procedure in itself. We find such solo droplets because of the random nature of cell encapsulation in droplets, which follows a Poisson distribution (Supplementary Fig. 2). Since the cell counting procedure breaks the droplet emulsion, the total counts by flow cytometry are a mixture of cells liberated from true mix droplets and solo droplets. Imaging after 24 h indicated that ca. 26% and 23% of co-culture droplets were occupied solely by either *P. putida* or *P. veronii*, respectively (*solo* droplets), and 26% contained both (*mix* droplets; the other 25% being empty). The increased proportion of *P. veronii* in the fused co-culture droplet emulsions counted by flow cytometry is thus increased by the fraction of *P. veronii* solo droplets. True co-culture droplets containing both *P. putida* and *P. veronii* showed an average 11.3% reduction in median productivity of *P. putida* (Fig. 2e, *mix*, $p = 0.0056$ in t-test to solo droplet productivity, $n = 3$), which is similar as measured by flow cytometry counting on fused droplets (Fig. 2d). Productivity of *P. veronii* was indifferent between *solo* droplets (in co-culture) and *P. veronii* mono-culture droplets (Fig. 2e, $p = 0.5073$, two-sided t-test), but—as expected, was on average 79.4% inferior in droplets with *P. putida* present (Fig. 2e, $p = 5.68 \times 10^{-4}$, two-sided t-test). Mix droplet outcomes effectively ranged from those with almost exclusively *P. putida* to almost exclusively *P. veronii* and some with more equal proportions (Fig. 2f). Interestingly, ca. 24% of the co-culture droplets occupied with both

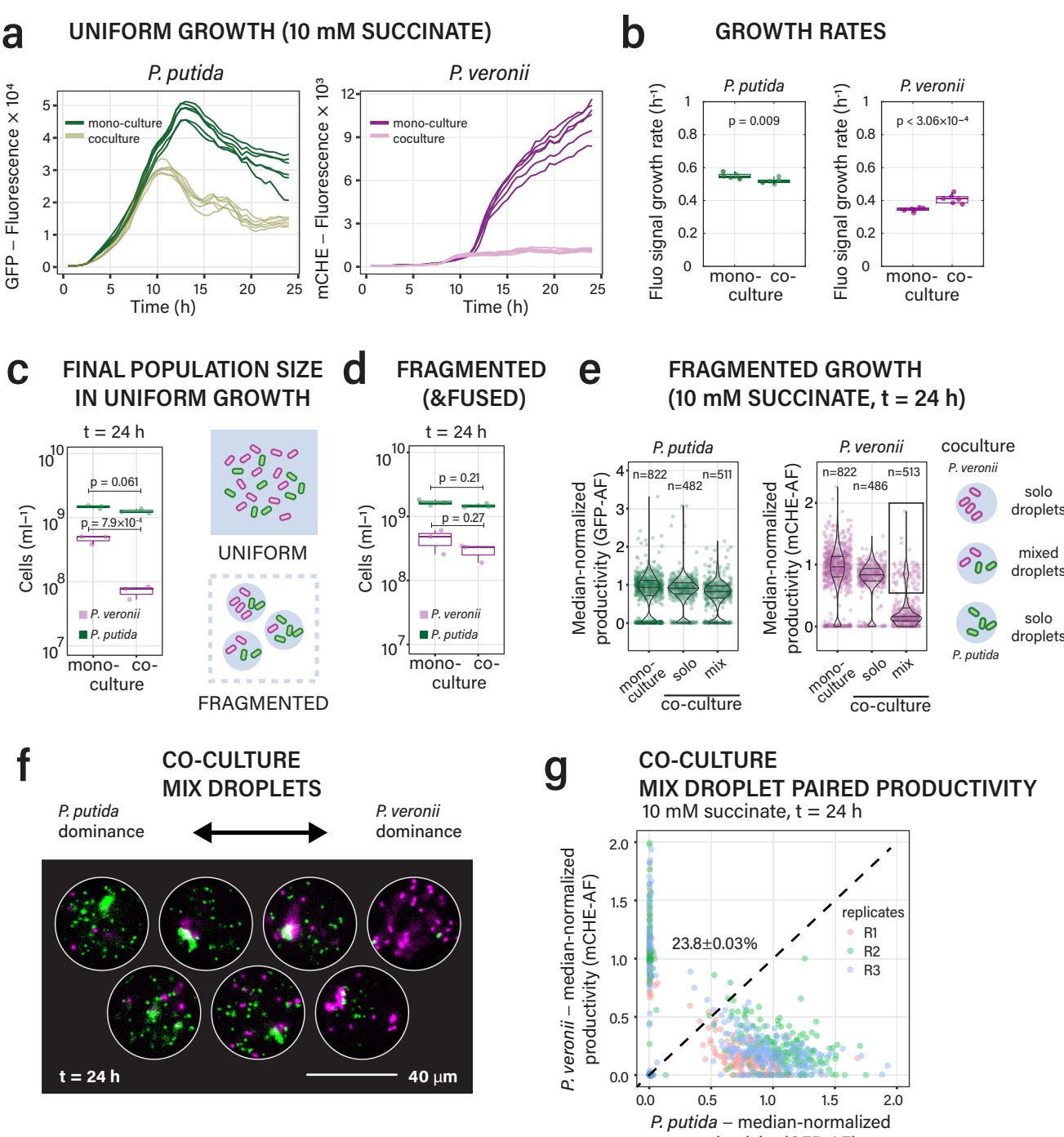

**Fig. 2 | Competitive growth of *P. putida* and *P. veronii* on a single shared substrate in suspended and fragmented culture. a** Comparative growth of *P. putida* and *P. veronii* in uniform suspended culture, alone (mono-) or in co-culture, measured by the increase of total strain-specific fluorescence. For stationary phase mono-culture productivities from culture turbidity, see Supplementary Fig. 3. **b** Inferred maximum specific population growth rates ($n = 6$ replicates). Box plots show the median, lower and upper quartiles, and individual data points. The whiskers extend to the most extreme data points not considered outliers. *P*-values from two-sided *t*-test. **c** as (**b**) but for the initial population size (as cells mL$^{-1}$) after 24 h in uniform liquid suspension, or in suspension from coalesced droplet emulsions (**d**), measured by flow cytometry, for mono- and co-cultures of *P. putida* and *P. veronii*. Each dot is the mean count of a biological replicate ($n = 3$). *P*-values from two-sided *t*-tests. **e** Per droplet productivity in fragmented growth conditions after

24 h (violin plot with mean, upper and lower quartiles, and individual droplet values, *n* number of imaged droplets). For comparison to the starting situation, see Supplementary Fig. 4. Note the distinction between mono-culture droplets and solo droplets (i.e., those droplets in a co-culture occupied by chance by only a single strain). Black frame on the right panel highlights mixed droplets where *P. veronii* reaches productivities similar to mono-cultures. **f** Visual impression of the gradient of paired productivities, yielding dominance of either one of the strains. **g** Median-normalized paired productivities after 24 h in droplets with both strains ($n = 3$ independent replicate droplet incubations). Dotted line indicates suggested equal biomass of both partners, showing dominance of *P. putida* growth but a significant proportion (23.8%) of inversed competitive outcomes (i.e., *P. veronii* dominance). Source data are provided as source data file.

strains were dominated by *P. veronii* (Fig. 2g), which may be due to non-growing cells of *P. putida* (Supplementary Fig. 4) or to a competitive gain by *P. veronii* in presence of growing *P. putida* cells. Control experiments with paired isogenic *P. putida* each expressing a different fluorescent protein, showed an equilibrated distribution of mix droplet outcomes under succinate competition (Supplementary Fig. 5), confirming that the observed outcomes of mix droplets with *P. veronii* and *P. putida* are due to kinetic and phenotypic differences among the strains. This result indicated, therefore, that in fragmented growth conditions with low founder cell densities, *P. veronii* can overcome its general competitive disadvantage for growth on succinate.

## Fragmented growth habitats yield varying outcomes even in case of substrate independence

To contrast the reversion of competition outcomes, we next imposed a substrate independence scenario, in which *P. putida* and *P. veronii* were each given an exclusive carbon substrate (Fig. 1a). Our expectation here was that since substrate competition would be alleviated, both strains would grow unhindered, and liquid-suspended and pL-droplet growth would be largely similar. To test this, we used a previous observation that *P. putida* consumes putrescine but not D-mannitol, whereas *P. veronii* prefers D-mannitol and only very slowly metabolizes putrescine[51]. Indeed, in this case, the measured growth rates in uniform liquid suspension were indifferent between mono- and co-culture conditions for both *P. putida* and *P. veronii* (Fig. 3a, $p = 0.6300$, $p = 0.3990$, $n = 6$; growth curves in Supplementary Fig. 6), although the time until first doubling was around 20% shorter for *P. putida* in co-culture (Fig. 3b, $p = 0.0032$). Also, the total productivity (in cells mL$^{-1}$ determined by flow cytometry) was unchanged between mono- and co-cultures, for both *P. putida* and *P. veronii* (Fig. 3c, $p = 0.84$, $p = 0.46$). In contrast, the total productivity in fragmented conditions was two-fold higher for *P. putida* than *P. veronii*, but again indifferent between mono- and co-culture conditions (Fig. 3c). Seen at population levels, these results thus suggested substrate independence for either species in uniform liquid suspended and pL-droplet growth.

At the level of individual co-culture mix droplets (i.e., having detectable fluorescence signals of both *P. putida* and *P. veronii*), the substrate independence scenario presented itself again very differently. An average of 5.5% of droplets were dominated by *P. veronii* (Fig. 3d, fraction **a**), whereas 39.2% consisted of droplets where productivities were equal (Fig. 3d, fraction **b**, $n = 4$ biological replicates). In contrast, 48.7% of co-culture droplets were largely dominated by *P. putida* (Fig. 3d, fraction **d**). Partly, the outlier fractions **a** and **d** may again be due to incidental non-growing cells of either partner (Supplementary Fig. 4). The median productivity of *P. putida* was higher in the fraction **b** droplets (Fig. 3e, ANOVA, post-hoc $p = 0.0213$, compared to *P. putida* solo droplets), whereas that of *P. veronii* in fraction **b** was lower compared to *P. veronii* mono-culture droplets (Fig. 3e, $p = 5.67 \times 10^{-4}$; $n = 4$ replicates). Compared to a *null* model of co-culture droplet distributions, the productivity of *P. putida* was indeed significantly higher in mix droplets with *P. veronii*, but significantly lower when being in droplets alone, than expected from the sampled probability of its individual productivities in mono-cultures (Fig. 3f, ANOVA with post-hoc test $p$-values are 0.0492 and 0.0003, grid fractions **e** and **d**, respectively). This indicated that interactions in droplets with equal-sized partner populations were mutualistic for *P. putida* and slightly antagonistic for *P. veronii*. These results thus illustrate how a globally perceived non-competitive scenario breaks down in a variety of different outcomes in a fragmented habitat.

## Fragmented growth effects under an inhibition scenario

To explore whether fragmented growth impacts variable outcomes in situations beyond substrate competition, we used two further strain combinations, which illustrate an inhibition and a killing interaction.

In the first of these, we produced an inhibition scenario, consisting of a phyllosphere isolate *Pseudomonas* sp. Leaf15, known to excrete a growth-inhibitory compound[48], mixed with a sensitive strain (for which we used a fluorescently tagged variant of *S. wittichii* RW1[49]). In this scenario both strains have their own carbon substrate, to avoid generating additional substrate competition (Fig. 1a). We used succinate for L15, which is not measurably used by RW1, and salicylate for RW1, which is not used by nor toxic for L15 (Supplementary Fig. 7). As expected, growth rates of RW1 in co-culture uniform liquid suspension with L15 were reduced by 25% compared to its mono-culture, whereas those of L15 are unaffected, confirming growth rate inhibition (Fig. 4a, b). Despite the growth rate decrease, the final attained population size of both RW1 and L15 in uniform liquid suspension co-culture was indifferent from the mono-cultures (Fig. 4c, measured by flow cytometry; $p = 0.95$, $p = 0.79$). Also, the productivity of RW1 in mix droplets with L15 was similar to that in solo droplets (Fig. 4d, $p = 0.13$, $p = 0.53$, $n = 4$; Supplementary Fig. 8), although both showed a constant ca. 10% fraction of non- or poorly growing cells (Fig. 4e). Compared to mono-culture growth, productivity of RW1 was the same and that of L15 slightly higher in co-culture mix droplets ($p = 0.0020$; sign rank test on median of the growing droplet fraction, Fig. 4f, Supplementary Table 1). However, there was a 0.8–9.3% (average 3.2%) fraction of mix droplets with RW1 and L15 productivity higher than expected from their mono-culture droplet growth (Fig. 4f, f2 fraction, $p = 0.0039$, sign rank test all time points and replicates). This fraction thus represents local positive interactions, suggesting reversal of inhibition under fragmented growth conditions.

## Tailocins provide a competitive advantage only within fragmented habitats

In the final example, we studied the interactions between two *P. protegens* strains, one of which (Pf-5) is sensitive to a phage tail-like weapon, or tailocin, produced by the other (CHA0), leading to its lysis[50] (Fig. 1a). CHA0 is self-resistant to its own tailocins[50]. Activation of tailocin production and release, however, is a rare event in CHA0 cultures and requires a stress trigger[50]. Consequently, we expected that variable tailocin production may occasionally change the competitive outcome during growth on the same substrate, which would be detectable under fragmented growth, but not in uniform liquid suspended cultures.

Co-cultured strains on a single common substrate (succinate) in uniform liquid suspension indeed yielded almost equivalent substrate competition outcomes, with equal time to reach stationary phase for both CHA0 and Pf-5 in mono-culture, and ca. 50/50 yields in stationary phase (Fig. 5a). Productivities of either strain in co-culture pL-droplets were also equal and approximately half of that in mono-culture droplets (Fig. 5b, solo). The observed distribution of the productivities of Pf-5 and CHA0 in mix droplets followed an almost perfect constant sum, composed of the variation of individual productivities of Pf-5 and CHA0 (Fig. 5c).

Interestingly, however, in a small fraction of individual droplets with both CHA0 and Pf-5, an increased background fluorescence in the mScarlet-I fluorescence channel for Pf-5 could be observed, which in timelapse droplet imaging appeared as sudden onsets of partial and even complete disappearance of Pf-5 cells (Fig. 5d, Supplementary Movie 1 and 2). This sudden disappearance of Pf-5 cells would be in agreement with the release of tailocins from CHA0 leading to the puncturing and liberation of the cell content of the sensitive Pf-5 cells (consequently leading to an elevated background fluorescence by diffusion of mScarlet-I protein). From the variation of Pf-5 median background fluorescence in solo droplets (Fig. 5e), we estimated that on average ca. 0.5% of all droplets with both partners show evidence for lysis of Pf-5 (i.e., above 2.5× the Pf-5 solo background standard deviation, Fig. 5e; Fig. 5f, $p = 0.0114$, $n = 3$ biological replicates, two time points combined). In summary, these results indicate that both *P.*

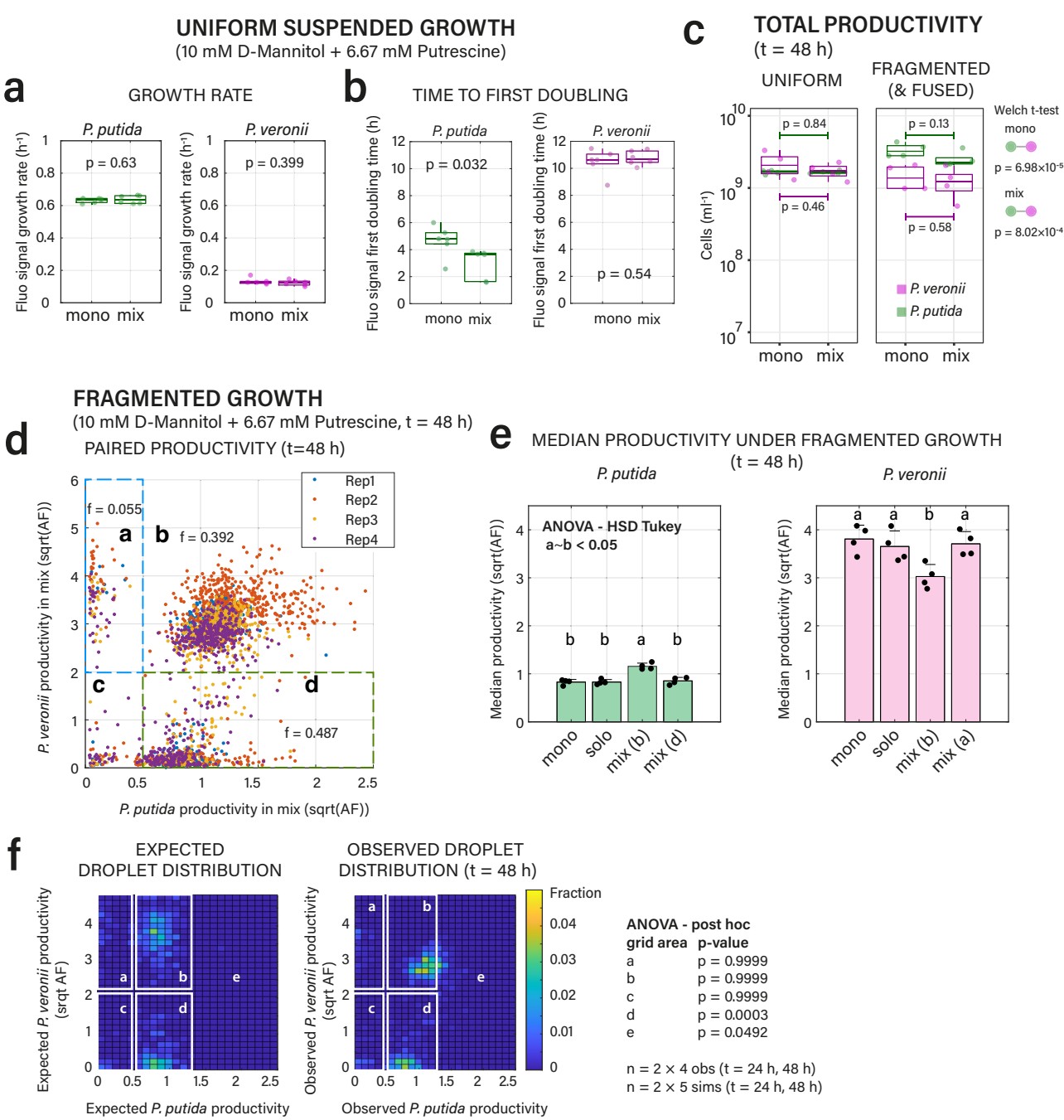

**Fig. 3 | Alternative growth outcomes in fragmented growth under substrate independence. a** Maximum specific population growth rates and (**b**) time to first doubling of mono- and co-cultures of *P. putida* and *P. veronii*, measured in 96-well plate liquid suspension in the presence of two substrates (prioritized by either of the strains: D-mannitol for *P. veronii* and putrescine for *P. putida*; *n* = 6 replicate values summarized by box plots with individual data points, *p*-value from two-sided *t*-test; growth curves in Supplementary Fig. 6). **c** Total population size (as cells mL⁻¹) measured by flow cytometry after 48 h for mono- and co-cultures of *P. putida* and *P. veronii* in liquid suspension (left panel) or in droplets (fused emulsions, right panel). Each dot is the mean count for each biological replicate (*n* = 4). *P*-values inside panels from two-sided *t*-test between mono and mix data. Welch two-sided *t*-test refers to fused droplet inter-species comparison. **d** Paired productivities after 48 h of *P. putida* and *P. veronii* in mixed droplets, overlaid from four independent biological replicates (REP1–4, individual colours), cultured on D-mannitol and

putrescine (square-root-transformed AF-values). Fractions (**f**) denote the mean values of the proportion of paired droplet productivities falling in the grid areas labelled (**a–d**)(separated by dotted lines). **e** Increased median productivities (bars, ± one *SD*) for *P. putida* (left panel) but not for *P. veronii* (right panel) in mix droplets area *b*, compared to mono, solo and mix-area *d* droplets. *P*-values on median productivities by ANOVA coupled with two-sided *post-hoc* HSD Tukey test (*n* = 4 biological replicates, lettering indicating significance groups at *p* < 0.05). **f** *P. putida* grows better than expected in mix droplets with *P. veronii* (seen here as shift into grid area *e*). *p*-values from ANOVA two-sided *post-hoc* multiple test, with *n* = 4 biological replicates and *n* = 5 simulations, combined from two time points. Expected droplet distribution for pairs simulated from individual mono-culture droplet distributions at the same time point (assuming no interactions). Source data are provided as source data file.

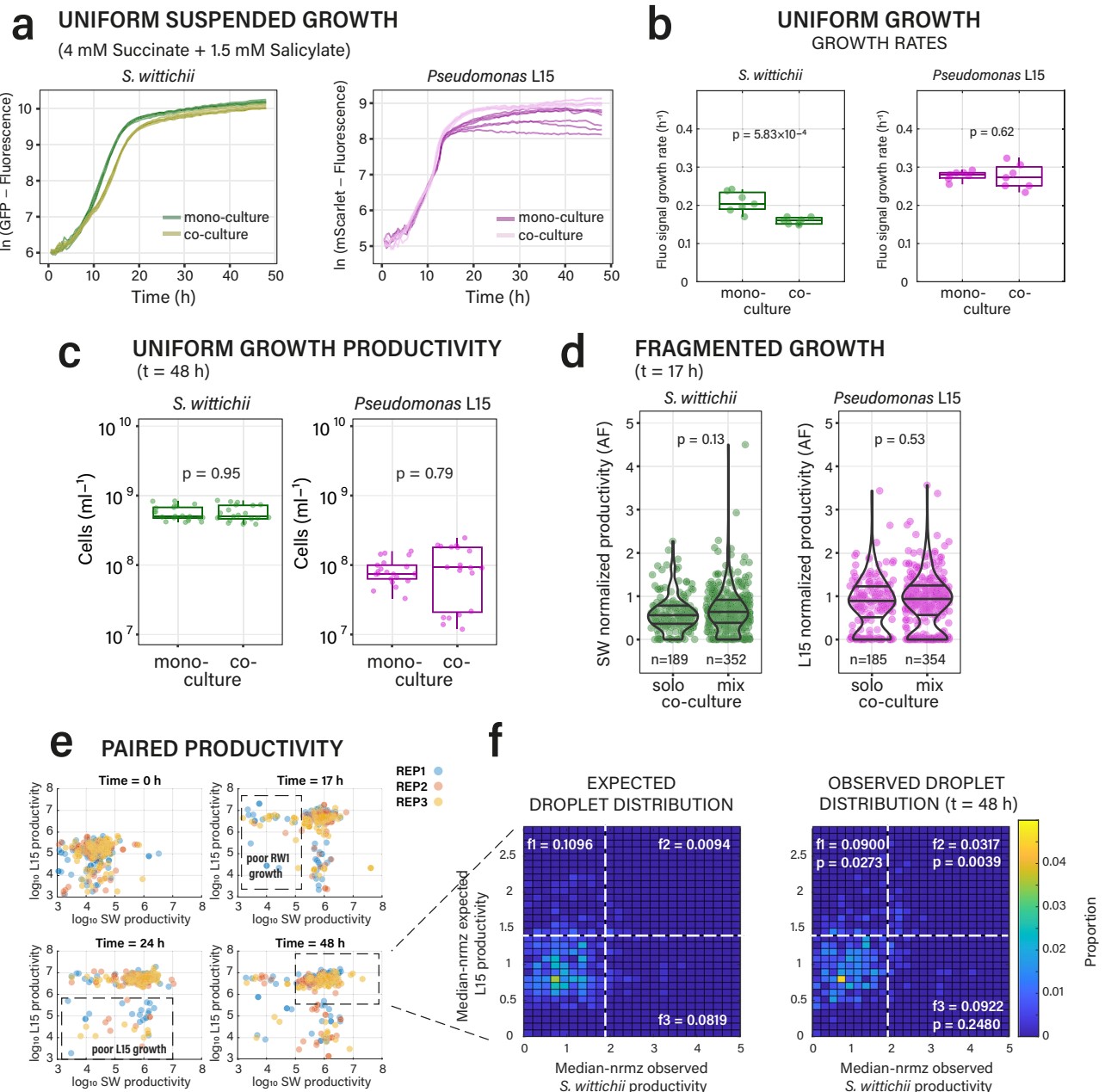

**Fig. 4 | Fragmented growth can lead to overturning of inhibition under substrate independence. a** Growth of *S. wittichii* RW1 (SW) and *Pseudomonas* Leaf15 (L15) cultured separately or together in 96 well-plates under substrate independence (*n* = 7, measured from strain-specific fluorescence). For growth on individual substrates, see Supplementary Fig. 7. **b** Maximum specific population growth rates in suspended growth conditions from culture fluorescence measurements (*n* = 7; box plots show the median, lower and upper quartiles, and individual data points, with whiskers extending to the most extreme data points not considered outliers). *P*-values from paired two-sided Wilcoxon rank-sum tests. **c** Total productivity (as cells mL⁻¹) measured by flow cytometry after 48 h in suspended culture (n = 3 biological replicates with each 7 technical replicates). *P*-values from two-sided *t*-tests between mean counts of biological replicates. **d** Per droplet productivities of either strain in solo and mix fragmented culture at *t* = 17 h (*n* = number of observed droplets; normalized to the median of the solo droplets at *t* = 48 h). *P*-values from

two-sided Wilcoxon rank sum tests. **e** Strain-specific productivities over time in mix droplets (log₁₀-transformed AF values, three replicates overlaid, each subsampled to 100 droplets), indicating the ca. 10% of droplets with poor growth of either partner. **f** Comparison of expected vs observed growth of pairs, removing the poorly-growing fractions (i.e., enlarged boxed region in (**e**)). AF-values normalized by their distribution maximum (roughly equivalent to the median). Paired distributions arbitrarily separated to fractions above or below average productivity properties to facilitate comparison (*n* = 5 simulations for each time point and observation replicate, *n* = 3 replicate observations, three time points). *P*-values from one-sided Wilcoxon sign-rank test including the same grid fraction across all replicates and time points (*f*-values denote their mean). Expected paired droplet distributions simulated from observed solo droplet growth (assuming no interactions). Source data are provided as source data file.

*protegens* strains are equally competitive for succinate, but that the production of tailocins by CHA0 can help to remove the competitor. Tailocins can thus have a crucial localized effect in co-inhabited microhabitats, but this effect is masked in uniform liquid-suspended culture, because of their low activation rate.

**Phenotypic variation in growth kinetics of founder cells determines colonization outcomes in microhabitats**

Since all the co-culture outcomes under fragmented conditions showed important variability compared to well-mixed uniform bulk conditions, we wondered if this would be the result of inherent

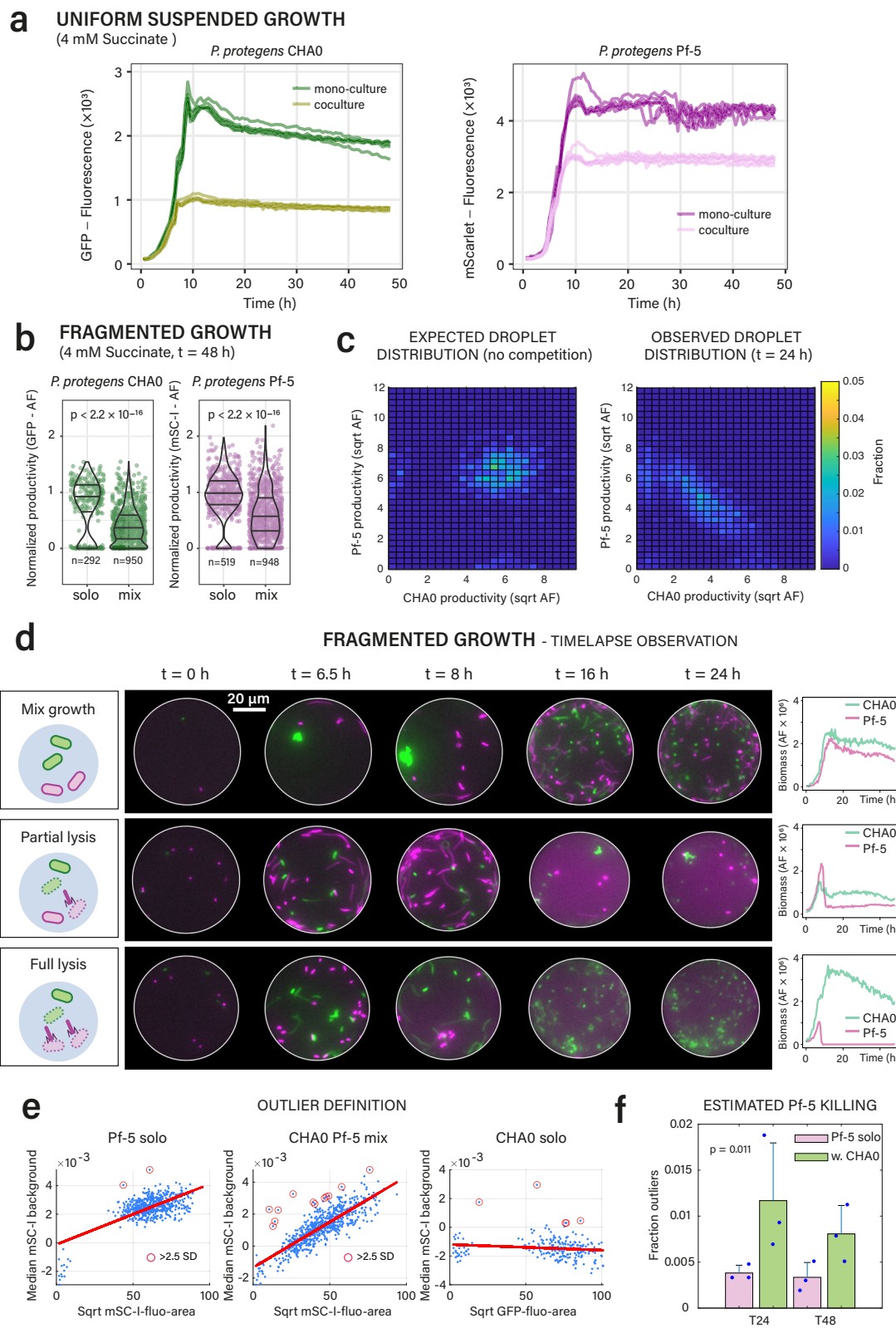

founder cell phenotypic variability. To demonstrate this, we again focused on *P. putida* and *P. veronii* and a single competitive substrate, and measured growth in individual droplets over time. For this, we used microfluidic chips with a low ceiling (10 μm height)[52], so that droplets are squeezed, kept in place and more cells are in perfect focus. Although the incubation in PDMS-glass results in slightly different oxygen provision to growing cells than culturing them in a pL-

droplet emulsion, it enabled measuring the variability of cell growth in individual droplets (Fig. 6a). Indeed, timelapse imaging confirmed different outcomes from the same starting configurations (e.g., one *P. putida* cell and one *P. veronii*, Fig. 6b, at *t* = 0 h), and growth measurements of *n* = 191 individual droplets showed kinetic variability both in mono- and co-culture droplets (Fig. 6c). Average growth rates of *P. putida* in solo droplets were 1.2 times higher than in mixture,

**Fig. 5 | Heterogeneous tailocin production ensures infrequent competitive dominance in microhabitats. a** Equivalent growth of *P. protegens* CHA0 and Pf-5 cultivated separately or together with 4 mM succinate in liquid suspension (*n* = 7 replicates). **b** Per droplet productivities of both strains in solo and mix droplets after 48 h (normalized to the median of the solo droplets). *P*-values from two-sided Wilcoxon rank-sum tests. **c** Expected (from solo droplets, without competition) and observed mix droplet productivities after 24 h, indicating almost equal competitive substrate sharing (square-root transformed AF-values). **d** Effect of tailocin lysis in individual droplets. Top: mixed growth without lysis; middle, partial lysis of Pf-5 (in magenta); bottom, total lysis of Pf-5 in the presence of CHA0. Time points selected from time-lapse imaging. Strain-specific fluorescence (biomass) development shown on the right. **e** Variation in mScarlet-I (Pf-5 solo droplets) fluorescent background sets outlier range, above which Pf-5 lysis is assumed (red-circled data points, >2.5 times standard deviation of the residual variation to the linear regression line). **f** Inferred fractions of Pf-5 lysis (bars show means ± one *SD*) in the presence of CHA0 compared to Pf-5 solo background (*n* = 3 replicates, *p*-value from two-sided *t*-test on combined *t* = 24 h and *t* = 48 h outlier fractions). Source data are provided as source data file.

whereas those of *P. veronii* remained indifferent between the two conditions (Fig. 6d). On average, *P. veronii* started dividing 4 h later than *P. putida* (Fig. 6e). Both strains also showed a tendency that incidental longer lag times decreased their final attained size in co-culture droplets (Supplementary Fig. 9). Paired growth trajectories were highly variable between droplets, even under the same starting cell-census (Fig. 6f and Supplementary Fig. 10), whereas unequal starting cell ratios tended to favour either one of the strains (Fig. 6g). However, growth rates and lag times of *P. veronii* were the only significant predictors for biomass ratio outcomes (generalized linear mixed effects model, $r^2 = 0.8236$, $n = 108$ co-culture droplet pairs, Fig. 6h, Supplementary Table 2), whereas founder cell numbers were less relevant. The variance in single droplet growth rates and lag times tended to decrease with increasing starting cell numbers (Fig. 6i, j, significant inequality of variances for *P. putida* but not for *P. veronii* - Brown-Forsythe test, see parameter distributions in Supplementary Fig. 11), suggesting that the influence of individual cell heterogeneities becomes less determinant and yields more averaged behaviour.

To better demonstrate the effect of single-cell growth variation on competitive outcomes in a two-species community within the fragmented habitat, we adapted an existing mathematical framework[51] for simulating carbon-limited competitive Monod growth of *P. putida* and *P. veronii* founder cells within 35 pL-droplets (Fig. 7a). In this simulation, each founder cell starts with independent growth kinetic parameters, subsampled from inferred distributions around means measured in liquid mono-cultures (Fig. 7a). In addition, each droplet is colonized by a Poisson-drawn random number of founder cells. Simulations including both individual growth kinetic variability and Poisson-distributed initial cell ratios ($\lambda = 3$ for both *P. putida* and *P. veronii*) produced growth distributions in co-culture droplets consistent with the observed distributions of *P. putida* and *P. veronii* co-culture droplet productivities (Fig. 7b, c). In contrast, both a constant starting cell number or homogeneous individual growth kinetics reduced the simulated fraction of droplets in which *P. veronii* reverses competition (Fig. 7d, e). Further simulations suggested then that mostly an increased heterogeneity of lag times and growth rates of *P. veronii* founder cells determines reversed competition outcomes with *P. putida* (Fig. 7f–h). This agrees with them being the most important predictors in the general linear mixed model outcome (Fig. 6h). Increasing kinetic heterogeneities among *P. putida* founder cells are also predicted to positively impact the reversal of competition by *P. veronii*, suggesting that *P. veronii* benefits from the heterogeneity among its competitor cells (Fig. 7f–h).

**Increased founder cell population sizes decrease competition reversal occurrences**

We suspected that the pronounced effect of individual cell phenotypic variation on the outcomes of paired strain growth at small founder populations would diminish with larger numbers. Simulations of *P. putida* and *P. veronii* growth under succinate competition indicated that increasing droplet volume and increasing starting cell numbers from 3, 10 to 100 per droplet (but the same starting cell density and thus the same number of generations of growth) resulted in more homogeneous outcomes (Fig. 8a). This occurred irrespectively of a Poisson-sampled or constant starting cell numbers per droplet

(Fig. 8a), and is mainly due to reduced variation at higher starting numbers (Fig. 8b). As a result, the proportion of *P. veronii*-dominated droplets decreases to almost undetectable at a starting number of 100 cells of each strain (Fig. 8a, inset). Seen across all droplets, this causes the relative abundance of *P. veronii* to diminish by 9 % from 3 to 100 starting cells per strain per droplet (Fig. 8c; for calculations not including solo species droplets, see Supplementary Fig. 12). Essentially the same effect is obtained when maintaining the same droplet volume (35 pL) but increasing starting cell densities (Fig. 8d, e). The difference here is that the final competitive relative abundance of *P. veronii* remains close to the initial species-mixing ratio of 50 %. The reason for this is the reduced number of generations that cells can undergo in droplets of the same volume but at higher starting numbers (Fig. 8f). Experimental reproduction of increasing starting cell numbers corroborated these simulations (Supplementary Fig. 13), resulting in reduced variation in droplet growth outcomes of competing *P. putida* and *P. veronii* (Fig. 8h, i), and a reduced proportion of *P. veronii* in the co-culture (Fig. 8j, Supplementary Fig. 14). Increasing cell densities in the same droplet volume then again increased the final ratio of *P. veronii* to *P. putida*, as expected from simulations and the number of generations of growth in the populations (Fig. 8j compared to the simulation of Fig. 8f, Supplementary Fig. 14).

Collectively, these results and simulations underscore that heterogeneity in growth properties among founder cells can enhance the probability of variable outcomes between competing species inhabiting fragmented habitats, particularly at low founder population census. This may also imply that some bacterial species have become selected for more variable growth kinetics, which can favour their survival under substrate competition in microhabitats.

## Discussion

Here we demonstrate how phenotypic variability in single-cell growth kinetic parameters and Poisson-variation in assembly of strain pairs in micro-scale communities lead to ecological outcomes that vary substantially from those inferred from global-scale interaction types. We show, using three different strain pairs and four different imposed growth regimes and interactions, how growth in fragmented environments at low starting cell densities (1–3 cells per droplet per species) enables local overturning of interaction directionality, whereas this is not detectable in uniform cultures starting at large population census (ca. $10^6$ cells). Simulations and experimental data further show that the effect of variation and overturning is most pronounced at starting numbers below 10 cells per droplet and species, above which it gradually diminishes (but does not disappear). This aspect of growth outcome variation and interaction reversal by habitat fragmentation has received little attention in previous studies describing paired species interactions. Even though the fraction of interaction reversals may seem relatively small, this effect can help to explain why less competitive species can locally sustain in mixed microbial communities within fragmented environments[53].

In all four tested different interaction scenarios and strain pairs, regular uniform liquid suspended culturing with larger volumes (here: 140 μL) and high starting cell numbers ($10^6$ cells) confirmed the intended global interaction types (i.e., substrate competition, independence, and growth rate inhibition). In contrast, fragmented growth

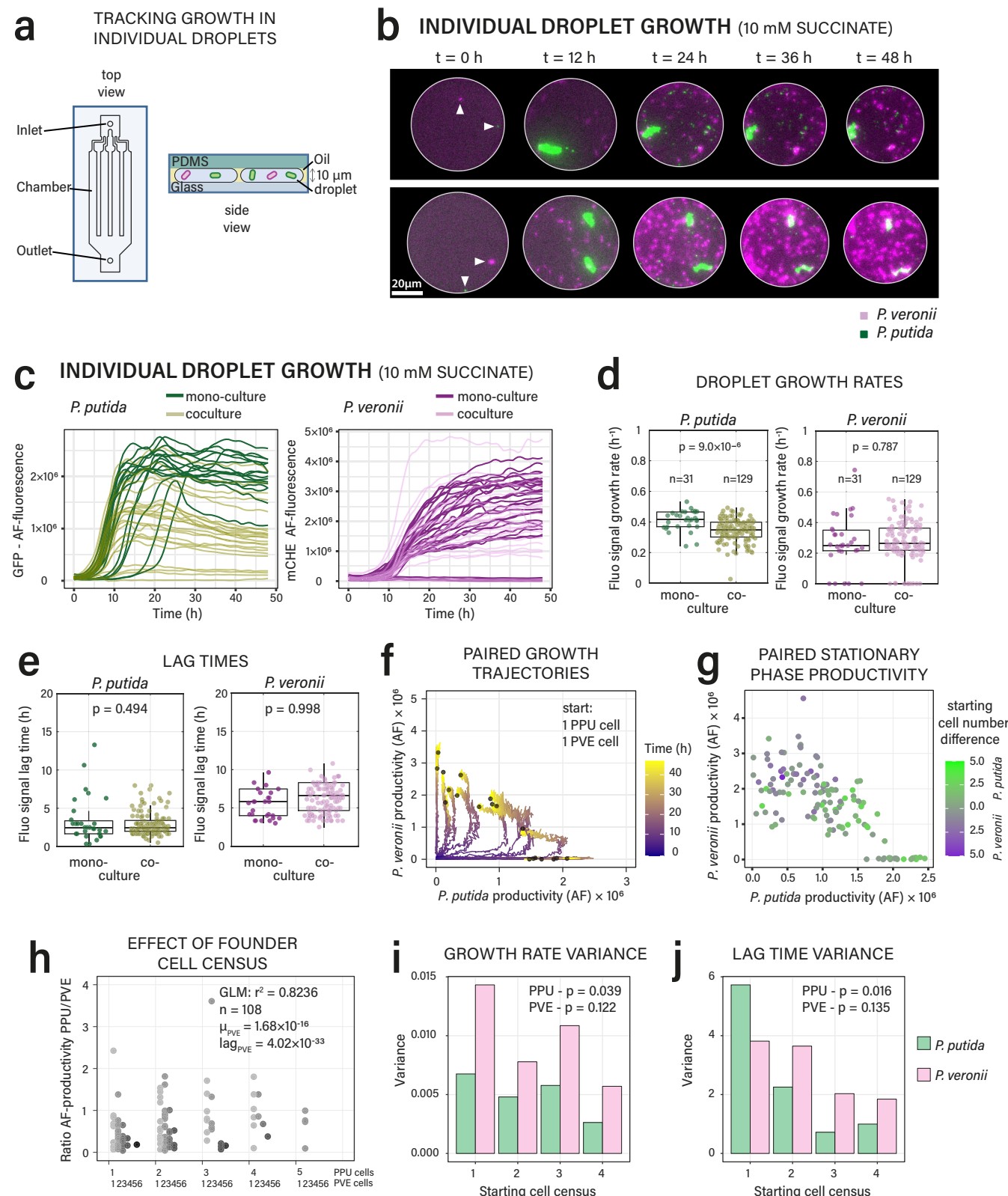

**a** TRACKING GROWTH IN INDIVIDUAL DROPLETS

**b** INDIVIDUAL DROPLET GROWTH (10 mM SUCCINATE)

■ *P. veronii*
■ *P. putida*

**c** INDIVIDUAL DROPLET GROWTH (10 mM SUCCINATE)

**d** DROPLET GROWTH RATES

**e** LAG TIMES

**f** PAIRED GROWTH TRAJECTORIES

**g** PAIRED STATIONARY PHASE PRODUCTIVITY

**h** EFFECT OF FOUNDER CELL CENSUS

**i** GROWTH RATE VARIANCE

**j** LAG TIME VARIANCE

of the same paired cultures led to variations in growth outcomes and interaction type reversal. For example, *P. veronii* grew to 4 times higher densities in fragmented co-culture droplets than under the same competition with *P. putida* in uniform liquid culture, and ca. 24% of mix droplets became dominated with *P. veronii* (Fig. 2). Global substrate independence, in contrast, led to the opposite: the appearance of individual droplets with higher-than-expected growth of either of the partners (Fig. 3). Global growth rate reduction on *S. wittichii* RW1 by *Pseudomonas sp.* Leaf15 was also inversed in 1–9% of isolated mix droplets, having higher than expected growth of the sensitive partner (Fig. 4). Since we prepare uniform liquid and droplet experiments in parallel with the same coculture suspensions having the same substrate concentrations, we do not a priori expect nor measured any difference in the extent of growth in bulk liquid (140 μl) and across all

**Fig. 6 | Heterogeneous growth kinetics and variable founder cell number determine competitive growth outcomes. a** Schematic of the observation chip used to track population growth within single droplets over time; figure panel redrawn after Fig. 1b from Ref. 52. **b** Time series examples of two droplets with the same number of founder cells for both *P. putida* and *P. veronii*, but with different outcomes. **c** Growth in individual droplets (each line a single droplet) of either strain in mono- or co-culture. Inferred maximum specific population growth rates (**d**) and lag times (**e**) in individual droplets of (**c**) (*n* = number of droplets with time-lapse observations). Averaged data is represented in box plots (median, upper and lower quartile, whiskers extending to the most extreme data points not considered outliers) with individual data points shown on top. *P*-values from two-sided *t*-testing. **f** Different growth trajectories of pairs with both a single founder cell, showing productivity of each member as its specific fluorescence (non-transformed AF-values). Black dots indicate final AF-productivities ($t = 48$ h). **g** Stationary phase per-strain productivity in mix droplets as a function of founder cell number (colours showing the difference in the number of starting cells, towards *P. putida* majority in green or *P. veronii* majority in magenta). **h** Effect of founder cell census on the ratio of stationary phase productivities (mean stationary AF-values) of *P. putida* (PPU) and *P. veronii* (PVE) in mix droplets. The initial number of both *P. putida* and *P. veronii* cells is indicated on the *x*-axis. Variation in the productivity ratio is best explained by variations in growth rates ($\mu_{PVE}$) and lag times ($lag_{PVE}$) of *P. veronii* (General linearized mixed effects model, $r^2 = 0.8236$, see Supplementary Table 2). Effect of founder cell census on the variance in the measured maximum population growth rates (**i**) and lag times (**j**) of either strain in droplets (mono and mix combined). *P*-values from Brown-Forsythe test. Source data are provided as source data file.

droplets (collapsed from the emulsion to a single suspension; Fig. 2). We conclude, therefore, that the observed variation in droplet community outcomes is not the result of some underlying confounding factor in growth conditions or cell-cell distances between a large and a miniature culturing system.

Our results do not only hold for interactions mediated by metabolic products but also for interactions involving bacterial killing by specialized weapons such as tailocins. Fragmented growth in isolated droplets showed that tailocin production by *P. protegens* CHA0 can eradicate *P. protegens* Pf-5 in a small proportion of microhabitats, whereas this killing has no effect in uniform bulk liquid cultures (Fig. 5). This small proportion is consistent with observed heterogenous activation of tailocin production at a population level, initiated in <1% of cells[50,54]. In addition, tailocins, like other specialized bacterial killing devices such as the type VI secretion system, act very locally[50], which has raised the question of their ecological importance[54]. Our droplet observations suggest that tailocins are particularly helpful in overturning competition in spatially restricted microhabitats. Within the context of the natural habitat of *P. protegens* (the plant rhizosphere), killing by tailocins could help to maintain local reservoirs of the producer strain, possibly increasing its survival and ability to colonize new habitats.

Two processes seem the most important to explain the effects of habitat fragmentation on microbial (paired) community growth outcomes: phenotypic variation of founder cells and stochastic sampling or dispersal effects on the formation and resulting species ratios in the microhabitats at start. At low starting cell numbers (1–3 per species per droplet), we find that phenotypic variation prevails over stochastic sampling effects (Figs. 6, 7), whereas at higher cell numbers (above 10 per species per droplet) the effects of both phenotypic variation and stochastic sampling on growth outcomes diminish (Fig. 8). Phenotypic variation is a well-known phenomenon from bacterial monocultures, caused by intrinsic and extrinsic molecular noise sources[55,56] that can affect different traits important for reproductive success, such as growth rate[57], lag phase[42], dormancy and antibiotic persistence[58,59], or metabolic specialization[60]. In some cases, for example, under influence of bistable genetic switches, phenotypic variation can lead to formation of subpopulations of cells with clearly different traits (e.g., sporulation[61], conjugation[62], virulence[63]). Under conditions where new habitats are colonized by large founder populations, such character trait variations will average out. But it can be easily conceived how, with founder populations of a few cells only, the reproductive success of the species in the pristine (i.e., newly colonizable) microhabitat in presence with others, is determined by the individual variation in cellular viability and traits. Our results demonstrate how phenotypic variation among founder cells, including incidental cell dormancy, is propagated into different growth outcomes (Fig. 6). In addition, variations in the species starting cell numbers through the processes of mixing and dispersal into the new habitat, will influence the probability of maintaining or averaging phenotypic variation in their starting populations, and thus determine their proliferation success in the microcommunity. This has important ecological consequences, as it may favour species coexistence.

Despite this general conception, the question is justified how relevant and representative are microhabitats with 35–200 pL and starting communities of 1–3 cells per species per droplet for microbial habitat fragmentation. An important premise for our work was to consider that natural environments for microbial communities are characterized by a high degree of spatial fragmentation and/or compartmentalization. Secondly, we assumed that such fragmentation and compartmentalization occur at a relevant micro-scale, such that the formed microhabitats are indeed colonized by dispersal of low numbers of founder cells and species. There is plentiful evidence to support the assumption that local habitats for prokaryotic cells measure in micrometer dimensions with low population census[22,64]. For example, an estimated 90% of microbial cell clusters in soils contain <100 cells[28]; plant surface architecture is characterized by micrometer crevices and microdroplets enabling microcolony formation[19,32], and sinking food particles in the ocean range in sizes from 1–50 μm with $10^2$–$10^3$ cells[65]. In addition, cell-cell interactions are assumed to be dominating at short (10–100 μm) distances[66,67], which is the cell-cell distance range attained within the confinement of single 40 μm droplets.

Although the exact number of founder bacterial cells in pristine environmental and host habitats is unknown, it likely ranges anywhere between a few and millions of cells, dependent on habitat and dispersal modes. For example, the most abundant microcluster sizes measured in soils[28] (with ca. 100 cells) are likely to have been seeded from fewer founder cells, and one can imagine how rainfall and subsequent drought cause re- and disconnection of soil pores, mixing small communities and enabling new local outgrowth. Studies of plant leaf colonization have shown that most aggregates in growth-favourable areas arise from single founder cells[68], which may be driven by the physics of microdrop formation resulting in solitary cells being enclosed[69]. Confocal scanning images of plant roots grown in soils inoculated with fluorescently tagged bacteria also show both solitary cells as well as small aggregates, suggesting de novo microcluster formation starting from single founder cells[70]. In contrast, colonization of the gastro-intestinal tract of human and animals is unlikely to occur from solitary bacterial cells, but rather from thousands to millions of cells simultaneously ingested with food particles. More generally speaking, the effect of founder cell population sizes on the variability of growth and interaction outcomes may thus be largely habitat-driven, but plentiful habitats for microorganisms seem to be colonized with low numbers of founder cells. Here, phenotypic and stochastic variation become ecologically relevant processes.

To measure the presumed effects of fragmentation on reproductive success, we relied on microfluidic pL-droplet formation and cell encapsulation. Droplet cultivation approaches have attracted interest as a high-throughput method to co-culture bacteria[15,37,38,52], or enrich bacteria from natural samples in an untargeted manner[71,72], and potentially allowing coculturing of unculturable members in multi-species conditions[73]. Notably, the droplet encapsulation creates

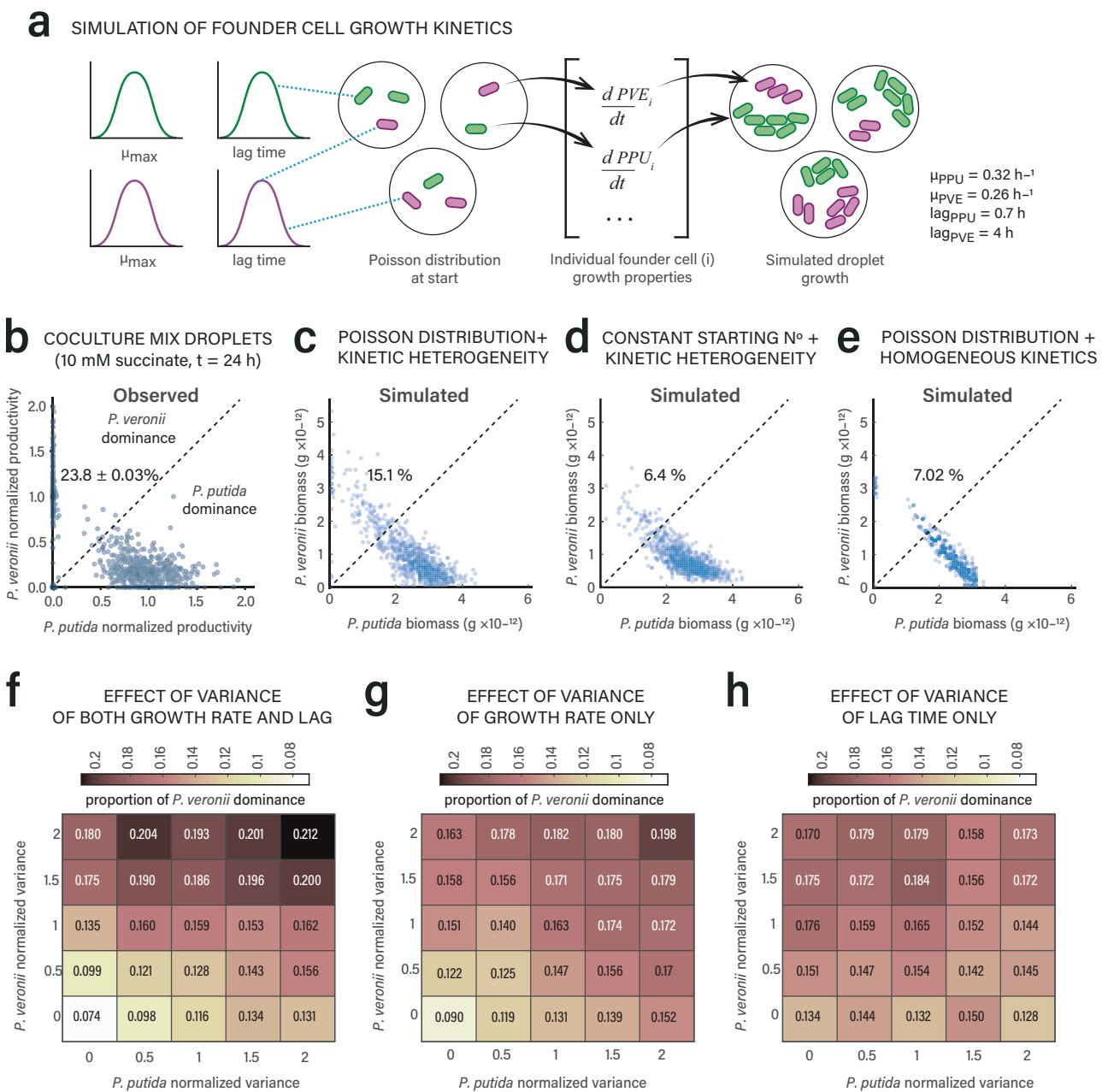

**Fig. 7 | Model simulations predict a positive effect of founder cell kinetic heterogeneities on competitive co-existence in fragmented habitats. a** Model workflow to simulate Monod substrate-limited (10 mM succinate) competitive growth of *P. putida* (PPU) and *P. veronii* (PVE) in droplets from Poisson-sampled individual founder cells, with individual subsampled growth rates and lag times (average values inferred from liquid suspended mono-culture growth, as in Supplementary Fig. 3). For model details, see *Methods* section. **b** Observed strain productivities of *P. putida* and *P. veronii* in mix droplets on 10 mM succinate after 24 h (reproduced from Fig. 2g for ease of comparison; normalized by the median strain-specific fluorescence of the droplet mono-cultures), in comparison to simulations with either both Poisson-distributed starting conditions and kinetic heterogeneity (**c**), only kinetic heterogeneity (**d**) or only Poisson-distributed founder cells (**e**). Percentages indicate the proportion of droplets with dominant *P. veronii* growth (i.e., either normalized fluorescence or simulated biomass above the diagonal trend line). Dots represent individual observed or simulated droplets (*n* = 3 replicates or simulations, 1000 droplets each). **f–h** Effects of variance in growth kinetic parameters of either strain on the proportion of simulated *P. veronii* dominance, either combined (growth rate and lag time) or individually. Normalized variance of 1 corresponds to the parameter values used in simulations of (**c**). Source data are provided as source data file.

---

isolated habitats that only allow local resource depletion and the development of metabolic or contact-dependent interactions, but no cross-talk between droplets[45]. The production of droplet encapsulation from species co-cultures leads to formation of empty and solo droplets (i.e., including only a single species member from the co-culture), which is dependent on the inoculum density. Droplet culturing, therefore, does not only enable observation of community interactions at individual droplet level, but extrapolation from the ensemble of all droplets helps to understand dispersal and fragmentation effects at the level of the meta-community. Subjecting communities to alternating cycles of droplet growth, collapse and mixing, and reseeding could be an interesting approach to study longer term ecological effects of microhabitat fragmentation.

Our findings help to explain why so many bacterial taxa with overlapping metabolic capacities but different growth rates can co-occur in the same macro-habitat, echoing similar conclusions from

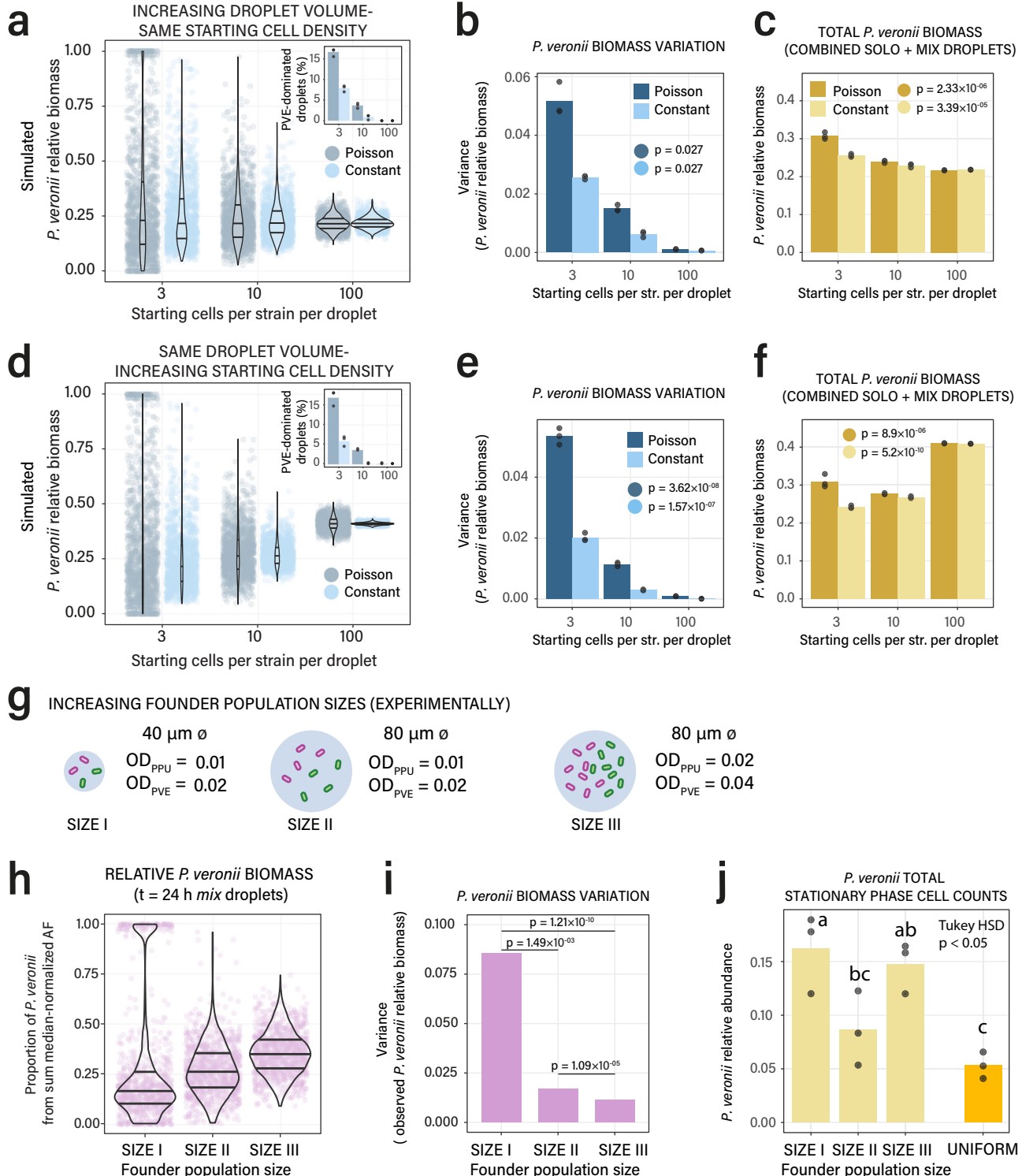

macro-ecology on spatial niche heterogeneities[74–76]. One can argue that slight differences in substrate utilization and metabolic dependencies may provide opportunities for co-existence[77]. In addition, the spatial isolation of habitats, and recurring processes of temporal mixing and dispersal can contribute to co-existence[20,78]. However, spatial fragmentation itself is not enough to maintain diversity if taxa would not show phenotypic variation, because, in the absence of cell-cell variability, interspecific interactions would become completely deterministic. We thus conclude that the importance of the micro-scale is not simply to provide spatial isolation, but to integrate variations in the dispersal processes leading to mixing of founder cells from

different species at low numbers[6], and varying emerging local inter-specific interactions as a result of individual cell phenotypes. In this light it would be reasonable to assume that there is selection on genotypes with wider phenotypic growth variation[41], as it could lead to increased chances of success to proliferate upon dispersal in mixed-species fragmented environments; a notion supported by our models (Fig. 7). Spatial fragmentation (or perhaps rather: dynamic variation in spatial fragmentation) thus plays a crucial role in types of local inter-actions and the resulting diversity of a complex meta-community[5,6]. To extrapolate downwards from globally measured interactions to small scales is not doing justice to the existing variability in such interactions

**Fig. 8 | Effects of increasing founder population sizes on droplet growth outcomes.** Simulated *P. veronii* relative biomass abundance (**a**) and variance (**b**) in droplet growth ($n = 500$ droplets, 3 replicate simulations, bars show the mean; Violin plot overlays show median, 25th and 75th percentiles) with increasing founder population sizes of both *P. veronii* and *P. putida* in proportionally increased droplet volumes (hence, same starting cell density). Two starting conditions: Poisson-sampled or constant cell numbers. Inset shows the proportion of *P. veronii*-dominated droplets (i.e., relative biomass > 0.5). *P*-values from two-sided Kruskal-Wallis rank sum test. **c** Total *P. veronii* relative biomass abundance compared to *P. putida* from solo and mix simulated droplets in (**a**). Bars show the mean overlaid with individual data points. *P*-values from ANOVA testing. **d–f** Simulations as in (**a–c**) but with constant droplet volume and increasing founder population sizes (hence: increasing starting cell densities). **g** Experimental strategy to achieve increasing founder cell population sizes and densities. OD, starting culture turbidities. PPU, *P. putida*; PVE, *P. veronii*. **h** Measured relative biomass of *P. veronii* (as the proportion of the *P. veronii* median-normalized AF signal compared to the sum median-normalized productivity per droplet of both *P. veronii* and *P. putida*) after 24 h incubation in the different droplet starting conditions (data points pooled from $n = 3$ replicate emulsions). **i** Decreasing variance in observed *P. veronii*-relative abundance at higher founder population sizes. *P*-values from non-parametric variance comparison Fligner test, adjusted by Bon Ferroni correction. **j** *P. veronii* stationary phase cell counts (by flow cytometry) as proportion of the sum of *P. veronii* and *P. putida* counts, after coalescing droplet emulsions to a single suspension, for the different founder population sizes ($n = 3$ replicates). Comparison: relative *P. veronii* cell counts in 96-well co-culture suspensions after 24 h, started with ca. 0.5 million cells per strain per 140 μl ($n = 3$ biological replicates). Letters indicate statistically significant groups at a *p*-value < 0.05 from an ANOVA with two-sided HSD Tukey *post hoc* test. Source data are provided as source data file.

and provides an oversimplification of their role in community development.

## Methods

### Strains, media and culture conditions

Two *Pseudomonas* strains were used for the substrate interaction experiments: *P. putida* F1 (PPU) is a benzene-, ethylbenzene- and toluene- (BTEX) degrading bacterium from a polluted creek[46]. *P. putida* F1 was tagged with a single-copy chromosomally inserted mini-Tn5 cassette carrying a constitutively expressed fusion of *eGFP* to the P$_{circ}$ promoter of ICE*clc*[79]. As isogenic control we used a *P. putida* F1 tagged with a single copy constitutively expressed mCherry from the *tac*-promoter. *P. veronii* 1YdBTEX2 (PVE), a BTEX-degrading strain, was isolated from contaminated soil in the Czech Republic[47]. PVE was tagged with a constitutively expressed *mCherry* from the P$_{tac}$-promoter within a single-copy inserted mini-Tn7 transposon (carrying a P$_{tac}$−*mCherry* cassette−as described in ref. [80]). For the growth inhibition experiment, we used *Pseudomonas sp* Leaf15 (L15), an antibiotic-producer isolated from *Arabidopsis thaliana*'s phyllosphere[48], and a fluorescently tagged version of *Sphingomonas wittichii* RW1[49]. L15 was tagged with a constitutively expressed *mScarlet-I* single-gene copy, using a pMRE-Tn7-145 mScarlet-I plasmid[81]. We used two rhizosphere-inhabiting strains of *Pseudomonas protegens*, CHA0 and Pf-5, for the tailocin interaction experiment. CHA0 was tagged with a constitutively expressed single inserted gene copy of *gfp2*[50], and Pf-5 was tagged with a constitutively expressed *mScarlet-I* from a single-copy inserted mini-Tn7 transposon (using a pUC18T-mini-Tn7T-Gm-Pc-mScarletI plasmid).

Strains were streaked on a nutrient agar plate directly from a −80 °C stock and were incubated for 2–3 days at 30 °C before being stored at 4 °C for the later experiments (max 12 days). Each biological replicate was started from a single isolated colony of each strain, which was resuspended in a Mc-Cartney glass tube with 5 mL (or 8 mL for L15 and RW1) of 21 C minimal medium (21 C MM, Supplementary Table 3, as described by Gerhard et al.[82]) supplemented with the appropriate carbon substrate(s). Cultures were incubated at 30 °C under rotary shaking at 180 rpm.

Precultures were centrifuged in 50-mL Falcon tubes to harvest the bacterial cells. The cells were two times successively washed in 5 mL of 21 C MM before being resuspended by pipetting in 5 mL of 21 C MM. *P. putida* and *P. veronii* cultures were centrifuged for 4 min at 12,000 rpm (Eppendorf centrifuge 5810 R with an F-34-6-38 rotor, 6 × 15/50 mL conical tubes), whereas the other four strains were centrifuged for 4 min at 8000 rpm. Specifically, the *S. wittichii* suspension was vortexed for 2 min at the final resuspension step to disperse cell aggregates as much as possible. The turbidity of the final cell suspension was measured with a spectrophotometer (MN−Nanocolor Vis, OD$_{600}$), and then diluted in 21 C MM with the appropriate C-source(s) to have approximately the same starting cell numbers (OD$_{600}$ of 0.02–0.05, depending on the strain; see Supplementary Table 4). For co-culture

experiments, the diluted cell suspensions of the respective partner strains (Supplementary Table 5) were mixed at a 1:1 ratio (*vol*/*vol*). Mono-culture controls were diluted two times with 21 C MM including the appropriate C-source(s), thus maintaining the same starting density for each strain in mono- and co-culture.

### Uniform liquid suspended growth in 96 well-plates

Aliquots of 140 μL of the freshly prepared mono- and co-culture cell suspensions were distributed in the wells of a 96 well-plate (Cyto One, tissue culture treated, Catalogue No: CC7682-7596), in six to seven technical replicates. Six to seven wells with the same sterile medium were incubated as controls for sterility. The plate was then incubated at 30 °C in a plate reader (BioTek, plate reader, Synergy H1), for up to 48 h. Plates were continuously shaken (double orbital, 282 cpm, slow orbital speed). Absorbance (OD$_{600}$) and fluorescence (GFP - 480/510 Ex/Em, and mCherry – 580/610 Ex/Em) were measured every 30 min in each cultivation well. After the incubation, the plates were placed on ice and sampled for flow-cytometry counting (see below).

### Microfluidic encapsulation of cells in droplets and culture procedure

The same prepared mono- and co-culture cell suspensions as for 96-well plate reader experiments were also used for microfluidic encapsulation (Fig. 1). This suspension contained on average $1.8 \times 10^7$ cells per mL, resulting in 0–3 founder cells at start within a 35 pL-droplet volume (examples of starting cell distributions presented in Supplementary Fig. 2). An aliquot of 500 μL of diluted cell suspension (mono- or co-culture) was taken up in a 1 mL syringe (Omnifix 1 mL, U-100 Insulin), and 1 mL of HFE 7500 Novec fluorinated oil containing 2% (w/w) of fluorosurfactant (RAN Biotechnologies, Inc.) was loaded in another one. Oil-dissolved surfactant stabilizes formed aqueous droplets and prevents them from coalescing. Syringes with the aqueous cell suspensions and with the oil were mounted on two separate syringe pumps (Harvard Apparatus, Pump 11 Elite / Pico Plus Elite OEM Syringe Pump Modules) to inject the liquids into a droplet maker microfluidic chip[44] at flow rates of 8 μL min$^{-1}$ and 20 μL min$^{-1}$, respectively. The droplet maker chip (custom-produced by Wunderlichip GmbH, CH-8037 Zürich, Switzerland) has a $40 \times 40 \times 40$ μm junction (see Supplementary Fig. 1), generating monodispersed droplets with a diameter of ca. 40 μm. Formed droplets were collected for 10 min in a 1.5 mL Eppendorf tube, corresponding to a total volume of 80 μL of droplets. The Eppendorf tube was prefilled with 250 μL phosphate-buffered saline (PBS) to prevent the droplets from evaporating. Eppendorf tubes with the droplets were kept on ice until being incubated at 30 °C to maintain the starting cell concentrations during the collection of the droplets from the different conditions tested in parallel (mono- and co-cultures). After a first timepoint imaging ($t = 0$ h), droplets were incubated at 30 °C and sampled at different intervals (Supplementary Table 5). After the final incubation time, the vials with the droplets were placed on ice before

coalescing all droplets into a single suspension and counting the resulting cell numbers by flow cytometry (see below).

## Testing effects of different founder cell population sizes

To specifically test the effect of increasing founder cell population sizes on competitive outcomes, we used again *P. veronii* and *P. putida* with 10 mM succinate under substrate competition but increased the junctions in the microfluidic device from 40 to $50 \times 50 \times 50\,\mu m$ and adjusted the oil-surfactant flow rate to $18\,\mu L\,min^{-1}$, to generate droplets of ca. $80\,\mu m$ diameter (ca. 268 pL volume). At preculture densities ($OD_{600}$) for *P. putida* of 0.01 and for *P. veronii* of 0.02, this yielded 2–6 cells of each species in $80\,\mu m$-droplets (measured distributions in Supplementary Fig. 13). By doubling the preculture densities, we obtained on average 3–8 cells per species in $80\,\mu m$-droplets (Supplementary Fig. 13). Droplet emulsions were then incubated and sampled as before.

## Droplet sampling

Droplet cultures were sampled at the start of the incubation, and after 17, 24 or 48 h (depending on the condition, Supplementary Table 5). An aliquot of $1.5\,\mu L$ was retrieved from the droplet emulsion, which was transferred by micro-pipette into a $5–\mu L$ HFE 7500 oil layer inside a chamber observation slide (Countess chamber slide, Invitrogen C10228). Then, another volume of $5\,\mu L$ of oil was added to the chamber to disperse droplets in a monolayer. Droplets were imaged at 3-5 random individual positions with a Leica DMi4000 inverted epifluorescence microscope (*P. putida - P. veronii* experiments) or a Nikon Ti2000 inverted epifluorescence microscope (for the two other paired-strain experiments), a Flash4 Hamamatsu camera, a 20x objective (Leica, HI PLAN I 20 x /0,30 PH1, with *P. putida-P. veronii*, or a Nikon CFI S Plan Fluor ELWD 20XC MRH08230, with the four other strains), in bright field (exposure time = 25 ms), red (exposure time = 400 ms) and green fluorescence (exposure time = 600 ms for *P. putida*, and 400 ms for the other strains). Images were collected as 16-bit.TIF files and further analyzed with a custom-made MATLAB script (vs. 2021b) to segment droplets and cells in droplets.

## Timelapse imaging of cell growth in droplets

In select droplet experiments (Supplementary Table 5), we followed the growth of cells in individual droplets over time by timelapse microscopy in an observation chip (Fig. 6a, chip design adopted from ref. 52, custom-produced by Wunderlichip GmbH). This polydimethylsiloxane (PDMS) print was directly mounted on a 1-well chambered Coverglass (Nunc™ Lab-Tek™ II Chambered Coverglass, Thermo Fisher, Cat number: 155360PK), to be able to immerse the chip during the observation. Before loading, the glass-bonded chips were placed in a vacuum chamber for 20 min to extract any gas contained in the PDMS, thus preventing the appearance of air bubbles during the incubation (we acknowledge that this potentially reduces the level of available oxygen to the cells). The chip was then filled and immersed in deionized filtered ($0.22\,\mu m$) water overnight. One hour before loading the droplets, the chip flow lines were emptied from the water and refilled with HFE 7500 oil, and the immersion chamber of the chip was filled with 1 mL of HFE 7500 oil, on top of which was placed 4.5 mL of deionized water, to limit oil and droplet evaporation during the incubation and imaging. Cell suspensions were encapsulated into droplets following the same procedure as explained above, but now the production chip outlet was directly connected by teflon tubing to the inlet of the (immersed) observation chip. An aliquot of $20\,\mu L$ of HFE 7500 oil was pipetted inside the observation chip inlet (using P20 tips) to allow good separation of the incoming droplets (this was done under live observation of the observation chip with an inverted microscope, to verify droplet separation). Droplets accidentally leaking into the chamber were removed by pipetting. Finally, aliquots of $30\,\mu L$ of HFE 7500 oil were pipetted onto the two outlets of the observation chip, to

prevent water from entering the chip during incubation and imaging. The immersion chamber was then closed and sealed with parafilm. The height of the chamber in the observation chip is $10\,\mu m$, which causes droplets to squeeze, and to almost completely fall within the focal depth range of the 20× objective. The chip was mounted on a Nikon Ti2000 inverted epifluorescence microscope with a programmable stage and was imaged every 10 min in the three channels (bright field, GFP and mCherry) as before, at the same individual positions set with the imaging control (Micromanager software 1.4.23). Images were exported as 16-bit.TIF files.

## Image analysis

TIF-images were processed in a custom-made MATLAB script (see Code availability), which segments all droplets per image and all fluorescent objects per droplet. The script then calculates the sum of all fluorescent objects per droplet (in pixel area), which is multiplied by their mean fluorescence intensity, to obtain a total fluorescent signal (area × fluorescence or AF, see Fig. 1). We use the AF-value per droplet as a proxy for the biomass production of the strain identified by its specific fluorescence (Supplementary Table 4), under the assumption that the more cells there are in a droplet, the higher their fluorescent signal will be (Fig. 6). We prefer using AF-values as biomass proxies instead of inferring per-droplet cell counts from AF-values or direct counting of cell objects, because of the potential variation in per-cell and growth-phase dependent expressed fluorescence, fluorescence distortion from cells out of the focusing plane, aggregation of cells into clumps, and cell movement during exposure time (e.g, Fig. 2F, Supplementary Movie 1 and 2). Depending on the scale of fluorescence intensities displayed by the cell-strain-pairs, the raw fluorescent signals were $log_{10}$-, square-root or median-transformed for display of potential subpopulations. In case of median-transformation, we use the mean of median AF-values from the corresponding mono-culture controls ($n = 3$ replicates) at the last time point (T24 or T48). The distributions of AF-signals are then analysed across all droplets, and across independent biological replicates. We consider droplets of co-culture experiments to be *solo* if only one of the fluorescence channels is detected, and otherwise a *mix* droplet (carrying cells from both encapsulated species). However, we made no a priori assumptions as to whether a founder cell is dormant or non-growing, which might be inferred from comparing T0 with T24 or T48 AF-values (Supplementary Fig. 4).

For timelapse experiments, images were segmented and processed in the same way as above, with the difference that a customized rolling ball algorithm was applied to compensate in the image segmentation for fluorescence variations existing among cells and bleaching of the signal over time. Additionally, the droplets were tracked between time frames, by comparing the distances of the centroid for every droplet between frame $t$ and the next frame $t + 1$. Droplets with minimum centroid distances were assumed to be the same on frame $t + 1$ as on frame $t$. Tracking of individual droplets was then manually controlled and corrected with the help of generated movies displaying the tracking ID attributed to each droplet over time. In this way, biomass development can be plotted per droplet over time, and the variation among droplets can be quantified.

## Fusing droplet emulsions for flow cytometry cell counting

Droplets from a single Eppendorf emulsion experiment were fused to produce a single aqueous phase, in which the total cell amount could be counted by flow cytometry. First, the extra HFE oil that settled below the droplet emulsion was removed by pipetting. To the remaining PBS and droplet emulsion layer, an approximate equivalent volume was added of HFE oil containing 1H,1H,2H2H-perfluoro-1-octanol (5 g solution Sigma-Aldrich, further diluted 4 times in HFE 7500 oil). This breaks the emulsion and fuses the droplets into a single

aqueous phase. The resulting droplet-cell-PBS aqueous phase was transferred into a new Eppendorf vial and its volume was measured from the micro-pipette directly.

## Flow cytometry counting of cell population sizes

Cell numbers in liquid suspensions from fused droplet emulsions or mixed liquid suspended cultures in 96-well plates, or precultures, were quantified by flow cytometry. Liquid cell suspensions were tenfold serially diluted in PBS (down to $10^{-3}$) and fixed by adding $NaN_3$ solution to a final concentration of $4\,g\,L^{-1}$ and incubating for max 1 day at 4 °C until flow-cytometry processing. Volumes of $20\,\mu L$ of fixed samples were aspired in a Novocyte flow-cytometer (Bucher Biotec, ACEA biosciences Inc.) at $14\,\mu L\,min^{-1}$. Events were collected above general thresholds of FSC = 500 and SSC = 150 to distinguish cells from particle noise, and gates were defined to selectively identify the strains from their fluorescent markers (Supplementary Table 4, see gating example in Supplementary Fig. 15). The Novocyte gives direct volumetric counts, which were corrected for the dilution. To convert cell counts from droplet suspensions to equivalent cell concentrations per mL, we considered the proportion of empty droplets from imaging and the extra volume of $250\,\mu L$ of PBS before droplet collection, as follows:

$$\frac{Cells}{ml} = \frac{Events}{20\mu l} \times 2000 \times 10^{Dilution\,factor} \times \frac{Droplet\,vol + PBS\,vol}{Droplet\,vol}$$
$$\times \frac{1}{Fraction\,of\,non\,empty\,droplets} \quad (1)$$

The multiplication by 2000 includes the 2-fold dilution when fixing the cells in the sample with $NaN_3$ solution, and the conversion to a per-mL concentration.

## Calculation of maximum specific growth rates, lag times and time to first population doubling

Average growth rates and lag times of strains in suspended liquid culture were inferred from the ln-transformed strain-specific fluorescence increase in mono-cultures grown in 21 C MM with their specific carbon substrate (as described above), each in 6–7 replicates. To average, we calculate the slope over a sliding window of five consecutive timepoints during the first 10 h, retained only slopes with a regression coefficient > 0.97; and reported the mean of those slopes as the maximum specific growth rate. Lag times were fitted from the complete (fluorescence) growth curve using a logistic function, and converted to *time to first population doubling* as the sum of the lag time (in h) plus the inverse of the logarithmic fitting constant multiplied by ln(2). In absence of lag time the time to first population doubling is the inverse of the maximum specific Monod growth rate.

To calculate growth rates from fluorescence in single droplets, we deployed a manual interactive plot of ln-transformed values of the summed fluorescence signal (the product of segmented area and the average strain-specific fluorescence in that area) over time, identifying the start and ends of the ln-linear range, and the *lag* time being the time between the start of the imaging series and the start of the ln-linear range. The maximum specific growth rate in the droplet was then taken as the slope over the entire identified ln-linear range. Since we did not segment individual cells, the summed fluorescence signal per droplet is a proxy for their biomass, and we report a Monod-type maximum specific growth rate.

## Mathematical model for population growth in droplets

We adapted a previously developed mathematical framework[51] to simulate the growth of *P. putida* and *P. veronii* populations in 35 pL droplets with nutrients (10 mM succinate). The initial resource concentration ($R_0$) is homogeneously distributed among all droplets and cannot diffuse between droplets. The chemical reactions inside each droplet are similar to the bulk population model in ref. 51, however, each founder cell follows its own differential growth equation, and includes possible kinetic variation. Growth of each founder cell $i$ in droplet $j$ thus follows the general reaction,

$$S_1 + R \xrightarrow{\kappa_{1_1}} P_1 \xrightarrow{\kappa_{1_2}} 2S_1 \,,\, P_1 \xrightarrow{\kappa_{1_3}} S_1 + W_1 \quad (2)$$

$$S_2 + R \xrightarrow{\kappa_{2_1}} P_2 \xrightarrow{\kappa_{2_2}} 2S_2 \,,\, P_2 \xrightarrow{\kappa_{2_3}} S_2 + W_2$$

where $S$ is the bacterial species, $R$ represents the resource, $P$ is the cell-resource intermediate state and $W$ any non-used metabolic side products. For simplicity, we did not consider cross-feeding effects. Each founder cell $i$ has its own lag time $L_{S_i}^{k}$ where $i \in 1, 2$ is the species index and $k \in \mathbb{N}$ the founder cell index. Therefore, the differential equations for species $S_1$ or $S_2$ present in droplet $j$ are,

$$\frac{dS_1(t)}{dt} = \sum_{i=1}^{N_{S_1}^{(j)}} \mathbb{1}_{\left\{t \geq L_{S_1}^{(i)}\right\}} \left( -\kappa_{1_1}^{(i)} S_1^{(i)}(t)R(t) + \left(2\kappa_{1_2}^{(i)} + \kappa_{1_3}^{(i)}\right)P_1^{(i)}(t)\right) \quad (3)$$

$$\frac{dS_2(t)}{dt} = \sum_{i=1}^{N_{S_2}^{(j)}} \mathbb{1}_{\left\{t \geq L_{S_2}^{(i)}\right\}} \left( -\kappa_{2_1}^{(i)} S_2^{(i)}(t)R(t) + \left(2\kappa_{2_2}^{(i)} + \kappa_{2_3}^{(i)}\right)P_2^{(i)}(t)\right)$$

$$\frac{dP_1(t)}{dt} = \sum_{i=1}^{N_{S_1}^{(j)}} \mathbb{1}_{\left\{t \geq L_{S_1}^{(i)}\right\}} \left( \kappa_{1_1}^{(i)} S_1^{(i)}(t)R(t) - \left(\kappa_{1_2}^{(i)} + \kappa_{1_3}^{(i)}\right)P_1^{(i)}(t)\right)$$

$$\frac{dP_2(t)}{dt} = \sum_{i=1}^{N_{S_2}^{(j)}} \mathbb{1}_{\left\{t \geq L_{S_2}^{(i)}\right\}} \left( \kappa_{2_1}^{(i)} S_2^{(i)}(t)R(t) - \left(\kappa_{2_2}^{(i)} + \kappa_{2_3}^{(i)}\right)P_2^{(i)}(t)\right)$$

$$\frac{dW_1(t)}{dt} = \sum_{i=1}^{N_{S_1}^{(j)}} \mathbb{1}_{\left\{t \geq L_{S_1}^{(i)}\right\}} \kappa_{1_3} P_1^{(i)}(t)$$

$$\frac{dW_2(t)}{dt} = \sum_{i=1}^{N_{S_2}^{(j)}} \mathbb{1}_{\left\{t \geq L_{S_2}^{(i)}\right\}} \kappa_{2_3} P_2^{(i)}(t)$$

$$\frac{dR(t)}{dt} = -\sum_{i=1}^{N_{S_1}^{(j)}} \mathbb{1}_{\left\{t \geq L_{S_1}^{(i)}\right\}} \kappa_{1_1} S_1^{(i)}(t)R(t) - \sum_{i=1}^{N_{S_2}^{(j)}} \mathbb{1}_{\left\{t \geq L_{S_2}^{(i)}\right\}} \kappa_{2_1} S_2^{(i)}(t)R(t)$$

where $N_{S_i}^{j} \in \mathbb{N}$ is the number of initial cells of species $i \in 1, 2$ in droplet $j$.

The starting number of cells of both species per droplet was drawn from a Poisson-distribution with an average of 3 cells per species. Individual growth rates and lag time parameters were sampled from a generated Gamma-distribution of *P. putida* and *P. veronii* growth parameters, inferred from mono-culture $OD_{600}$ curves with an Monte-Carlo Metropolis Hasting algorithm centred on the mean (method as described in ref. 51), and with variance deduced from the individual timelapse growth measurements in droplets with single-founder cells of *P. putida* and/or *P. veronii* (Supplementary Fig. 16). We also included a 15% chance for a cell to have a *zero*-growth rate, to account for growth-impaired cells that we observed from droplet imaging (Supplementary Fig. 4). Varying the heterogeneity in growth properties in Fig. 7f–h among founder cells thus consisted in increasing or decreasing the initial variance of the parameter gamma distributions.

## Statistical analysis and reproducibility

All experiments were carried out in biological triplicates (quadruplicate incubations for the substrate independence scenario). For each biological replicate, liquid-suspended cultures comprised 6-7 cultivation wells as technical replicates. Biological replicates of

fragmented droplet cultures comprised one separate emulsion incubation each, except in one of the replicates of the substrate competition experiment, for which a triplicate emulsion was generated, to assess and show the technical reproducibility of droplet cultivation experiments (Supplementary Fig. 17). Each emulsion sample was then imaged at 5–20 positions (technical replicates), to obtain 100–1000 droplets per mono- or co-culture and treatment. Flow cytometry counts (Figs. 2c, d, 3c and 4c) show the means of all technical replicates within each biological replicate. Each suspension from a cultivation well or fused droplet emulsion was counted three times by the flow cytometer, from which the mean was taken. T-tests were conducted to compare mean cell-counts in flow cytometry. Normality in the data was verified with a Shapiro-Wilk's test, and variance homogeneity was verified with a Fisher test. Median, top-10 or low-10 percentile productivities of each species in mono vs mix droplets were compared using Wilcoxon rank-sum or sign-rank tests (when taken across multiple time points). Depending on the data, we tested against a *null* hypothesis of sample means or ranked values being indifferent, or being higher or lower (i.e., a left or right tail). Tests were implemented in *R* version (within *R studio* version 2022.07.01) or in MATLAB (MathWorks, Inc. version R2021b).

To deduce strain interactions, we compared observed mixed droplet growth with the expected mixed growth from a *null* model based on probability distributions generated from the corresponding mono-culture droplet growth (i.e., assuming no interactions). The model uses the probability distributions for productivities of each of the strains in pairs at each sampled time point, simulated five times for the same number of pairs as the number of observed droplets. Expected and observed paired droplets were then counted in a productivity grid (e.g., as in Fig. 3f), and summed fractions across relevant grid regions (e.g., >1.5 times the median) were compared across replicates (typically, three biological replicates and five simulation replicates). *P*-values are derived from *ANOVA* comparison including all fractions, followed by a *post-hoc* multiple test (Fig. 3f), or by Wilcoxon sign-rank test in case of comparing multiple time points (e.g., Fig. 4f), as implemented in MATLAB.

To estimate the proportion of mixed droplets in which *P. protegens* Pf-5 might have been killed (lysed) by CHA0 tailocins, we deployed variations in the specific median fluorescence background originating from Pf-5. We first calculated the standard deviation in Pf-5 background fluorescence from Pf-5 solo droplets, corrected for the Pf-5 biomass (i.e., segmented area), which was multiplied by 2.5 as a boundary for the outlier range. This outlier range definition was then imposed on the Pf-5 specific fluorescence in mix droplets with CHA0 (and in CHA0 droplets where no Pf-5 area can be distinguished, assuming they may all be lysed). Outlier fractions were corrected for the total observed droplets and compared to the outlier fractions observed for Pf-5 solo droplets (i.e., as in Fig. 5e, f), using one- or two-tailed two-sample t-testing of replicate values.

The effect of founder cell census on the variance of growth kinetic parameters in strain-paired droplets was examined using a generalized linear mixed effect model (*glme*, as implemented in MATLAB 2021a), using measured maximum specific growth rates, lag times and starting cell ratios as variables. Droplets with quasi-null growth rates (which were also characterized by a lag time above 20 h) were removed for the analysis. The relationship of individual growth rate and lag time variance as a function of founder cells was further explored using a Brown-Forsythe test implemented in *R*, which tests the homogeneity of variances between groups without assuming the normality of the data.

## Reporting summary
Further information on research design is available in the Nature Portfolio Reporting Summary linked to this article.

## Data availability
Source data are provided with this paper. All raw imaging data, processed droplet productivity data, and numerical source data values underlying figure elements are available from a single downloadable link on Zenodo (https://zenodo.org/records/13342779) under accession 13342779 ([83]). Source data are provided with this paper.

## Code availability
All R/MATLAB scripts and code used for image extraction and analysis are reported and available from a single downloadable link on Zenodo (https://zenodo.org/records/13342779) under accession 13342779 ([83]).

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

## Acknowledgements
The authors thank Yolanda Schaerli and Florian Baier for their initial help in the droplet microchip setup, Vladimir Sentchilo for the genetic tagging of *Pseudomonas* strain L15, Bouke Bentvelsen and Guillaume Lieb for helping with image segmentation and Martin Ackermann for helpful discussions. This work was supported by the National Centre in Competence Research (NCCR) in Microbiomes (grant number 180575 to JM, CK and JAV).

## Author contributions
M.B., I.G., C.H., J.V. and J.M. conceived the studies and designed experiments. M.B. performed microdroplet and liquid-culture experiments. M.B. and J.M. analyzed droplet image data. H.T. wrote segmentation scripts for time-lapse data. I.G. wrote the simulation model. M.B. and I.G. conducted simulations. C.H., J.V., C.K. and J.A.V. provided biological material. M.B. and J.M. wrote the draft manuscript. All authors gave input, verified and corrected the written manuscript. C.K., J.A.V. and J.M. acquired funding and coordinated the work.

## Competing interests
The authors declare no competing interests.
