## [Peer Review File · Nature Communications]

Fragmented micro-growth habitats present opportunities for alternative competitive outcomesREVIEWER COMMENTS

Reviewer #1 (Remarks to the Author):

Summary:

I read the work of Batch and colleagues with pleasure. They use comparisons between batch and microdroplet co-culturing and combine this with theoretical modelling to study the effect of spatial fragmentation on bacterial interactions. They show that the fragmentation causes alternative outcomes than expected (based on batch culture) because of low-number fluctuations in the droplet inocula in combination with phenotypic variation within each co-cultured species.

The study is nicely quantitative, appears carefully done, and the manuscript is overall well written. The topic is timely, as understanding bacterial interactions is a key challenge to the field of bacterial communities, which play important environmental and medical roles. Their main finding is a valuable contribution worth publishing, but I have some reservations on the magnitude of the effect in natural environments.

Main:

**My main concern is the discrepancy between the small inoculum size in this study's model and experiments (3 cells) and typical inoculum sizes in natural systems, as based on literature reports (100-1000 cells).

The cell distribution across droplets follows a poisson distribution (LL158). Relative noise is often quantified as the standard deviation over the mean of the stochastic process. Poisson processes have a variance that is equal to the mean (λ), meaning that the relative noise scales as $1/\sqrt{\text{mean}}$. In other words, if $\lambda=3$ the relative noise is more than 50%. However, if $\lambda=100$ this decreases to 10% and with 1000 this is only 3%. That means that stochastic effects are expected to be much lower if the inoculum averages 100-1000 cells instead of 3 cells per droplet. The chance the underdog species gains stochastic numerical advantage decreases, and the effect of phenotypic variation reduces,

too, because of central limit theorem. This means that fraction of droplets with an inverted outcome (compared to bulk baseline expectation) diminishes.

Then the question is how large is a typical inoculating population? For soils, Ref. 27 is used to speculate that there are few cells per microhabitat (LL74-76), but it seems in this paper that the number of neighbours within interaction distance is about 100 cells, more in line with Ref 28 (LL341). Marine particles typically start sinking with 100-1000 founding cells (Ref 53). These numbers seem all much higher and a mismatch to the numbers used in this study. On the skin, Ref. 26 includes some modelling (their Fig. S16) that seem to suggest initial inoculation is low (e.g. 1) but that there is basically no competition in this case, as migration rates are very low. There have been some studies showing stochastically driven priority effects in for example flies (Obadia Curr Biol 2017) and worms (Vega and Gore, PLoS Biol 2017), that suggest host invasion is done by small populations (I do not know how small precisely), but this process is more complex than just fragmentation.

The authors should at least comment on this discrepancy. It seems easy to do some simulations to assess the magnitude of the effect with larger founding populations, and perhaps some additional experiments can be done with larger droplets and more cells per droplet.

**Then the comparison between batch and microdroplets appears somewhat convolved. This is because placing cells in microdroplets changes multiple things at once: effects of fragmentation in the sense of compartmentalisation (e.g spatial isolation) and effects of densities/spatial structuring (e.g average cell-cell distances).

As for the compartmentalization, the small volume means that only a modest population of 100 cells per droplet gives a bulk-equivalent concentration of $1E10$ cells/mL, meaning populations in droplets may reach steady state much faster (in approx 5 generations). It is therefore not clear if batch culture and droplets are in the same state after 24 hours. Or could the large variation even among solo droplets indicate the cultures have not reached a steady-state?

For the spatial organisation: 35 pL corresponds to a droplet diameter of 31 μm , suggesting typical cell cell distances of about $\sim 15 \mu\text{m}$. At 10^5 cells/mL, the founding concentration in batch culture, this is much larger, around 200 μm . To what extent does this distance change the interaction magnitudes? Apart from the killing story, which has an obvious distance effect, the authors do not really discuss the spatial aspect, while this actually generates more behavioral degrees of freedom in case of co-cultures. For example, a recent study has shown that species can change the spatial organisation of the interaction partner (e.g. D'Souza et al, PNAS 2023). I think this point can be addressed with mostly textual changes, by clearly distinguishing the two effects microdroplet compartmentalization causes, and clarifying which aspect the authors refer to in each case.

*The paper is overall well written but could be improved with some streamlining. The killing story seems somewhat like a separate story within this paper. The authors could move some material to the supplement to focus on the key ideas in the main text and figures.

Minor:

*The authors focus mostly on phenotypic variation of growth kinetic parameters, e.g the kind that is expected from stochastic gene expression (see Ackermann, Nat Rev Microb 2015 for a review). This is certainly interesting, but the population-level effects of such variation quickly averages out in populations larger than a 1000 cells (see above). Instead, I expect potentially much stronger effects of phenotypic variation that is bistable and heritable (for example the glycolysis-glycogenesis switch, Basan et al Nature 2020, or cell states such as bacterial dormancy), as these tend to preserve the variation to much higher cell numbers. Experiments with conditions and strains that show such large variations could be interesting to explore in future work, but perhaps the authors can already comment such scenarios.

L66: Ref.30 does show spatial organisation of the microbial community in the mouse gut, but I do not think from that can be concluded that there are microhabitats as this requires isolation.

L318: “This may also imply that some bacterial species have become selected for more variable growth kinetics “. It’s probably true that the mean value as well as the variation of growth rate are both subject to natural selection, but it is unclear why bacteria would involve a large variation in growth rate if they have the ability to just evolve a higher average growth rate.

Perhaps the authors can include infection biology in their list of examples with compartmentalization (as reviewed by for example Azimi et al, Nat Rev Microb 2022)

I think a very nice documented example of compartmentalization is the encapsulation by mucus in the mouse gut (Bergstrom, Science 2020). But I don't know how large the population sizes are in that case.

I think it would be nice to have cell number instead of the fluorescence signal when quantifying bacterial abundance of the two strains in each particle, I think the authors have the data to calibrate this (e.g. fluorescence per cell).

This is optional, but I was wondering if the authors could say anything on the current debate on the prevalence of positive vs negative interactions in microbial communities. Does fragmentation make competition or cooperation more likely?

Reviewer #2 (Remarks to the Author):

The work presented by the authors is a compelling and timely study of how phenotypic variation can determine the outcome of different microbial species interactions due to isolation in microhabitats. They use droplet microfluidics combined with classical microbial culturing approaches to tease apart the influence of standing phenotypic variation versus the “expected” outcome based on traits quantified from bulk culturing experiments such as the growth rate, lag-time, and yield. The authors test four scenarios:

direct competition, indifference, inhibition, and direct cell killing. In a significant fraction of the microdroplet incubators, the expected outcome of the final population composition is reversed. The authors further employ a stochastic model of population co-growth for the competition scenario fed with parameters inferred from the experimental results to test the relative importance of traits versus the stochastic phenotypic variation in the founder population. The authors conclude that this variation is a key driver of alternative interaction trajectories and can contribute to the maintenance of diversity.

This is a well-written potentially impactful manuscript that may benefit from some clarification and additional experiments/ simulations to disentangle the contribution of different mechanisms. Below, I outline some major comments followed by minor comments and figure-specific points. Overall, I am excited about this study which highlights the importance of studying microbial interactions and ecological principles at the scale that matters to individual cells in microhabitats.

Major comments:

1) The authors use combinations of different species and substrates to tease out the resulting population dynamics depending on competition, indifference, or antagonism (inhibition or killing). While this can highlight the importance of the standing phenotypic variation, it is confounded by the species-specific trait variations. As an interesting control, I would suggest using a single species (*P. putida* or *P. veronii*) tagged with two different fluorescent proteins and performing the same experiment in co-culture and fragmented growth. This would also provide excellent data to validate model predictions.

2) Although there is a section in the discussion devoted to the expected cell cluster sizes in e.g., soil and marine environments, the overall size of the droplets and cell numbers employed here are lower than what may be expected in natural environments. I believe there is an opportunity to use this experimental system (by increasing the droplet size or founder population) or the mathematical model (tuning the initial population size) to tease out if there is a threshold or simply a gradual change where species-specific kinetic parameters (the mean) becomes more important than the phenotypic heterogeneity in the founder population. Such an analysis will provide an important contribution to understanding both the importance of fragmentation in natural environments and also guide future experimental and engineering efforts when using droplet microfluidics. In

other words, can the authors derive a critical cluster size upon which selection overrides the influence of the initial standing variation?

3) When assessing the growth trajectories in individual droplets, a consistent pattern of rapid *P. putida* and lesser *P. veronii* productivity is observed, followed by a decline in *P. putida* productivity but a further increase in *P. veronii* productivity (Fig. 6E and Fig. S8). Depending on the recipe of the 21C minimal media (which I could not locate in the book reference of Gerhard et al. 1984 and only a PDF advert for the book is provided by PubMed when searching for the reference), *P. veronii* (and *S. wittichii*) may use alternative electron acceptors if the droplet experiences anoxia following rapid consumption by the growing population. Although this most likely does not influence the maximum observed growth rate, it can shift the balance in the final observed biomass within the droplets after 24 to 48h (especially at the high cell densities observed by the authors as shown in Fig. 2D).

4) One difficulty when assessing the results is the use of normalized fluorescence intensity to discern different conditions inside droplets. More specifically, I cannot understand from this data (e.g., Fig. 3D) if the droplets where *P. veronii* dominates are due to an initial droplet that did have both species but only *P. veronii* was able to grow (and *P. putida* was semi-dormant), or if *P. veronii* outcompetes the latter. A comparison of the normalized fluorescence of the founding community (media lacking an essential nutrient when creating the droplets or even including a growth inhibitor/antibiotic) and immediate quantification of the fluorescence could be an avenue to distinguish these cases (also for the bimodal distribution of normalized productivity in Fig. 2E, 4D, 5B, and 7B). Alternatively, since I believe images were taken at timepoint 0, these could also be used directly to establish a baseline fluorescence signal.

Minor comments (mostly details and phrasing):

Line 21: In my view it is more the “effect of being constrained within a microhabitat on the emergence of bacterial interspecific interactions” instead of the effect of a microhabitat.

Line 38: Some jargon is used without further explanation in the introduction which may diminish the readability, especially for a journal with a broad scope such as Nature Communication. Brief definitions for concepts common in microbial ecology such as founder cells or standing phenotypic variation would be appreciated.

Line 86: This sentence is difficult to understand. I would consider rephrasing to ensure that the meaning comes across as this is a critical sentence of the paper. I am not sure that microhabitats themselves favor the standing variation (or anything as a matter of fact, maybe promote? Or facilitate?).

Line 105: Closing bracket missing after the sentence.

Line 136: Is there a hypothesis of why *P. veronii* grew slightly faster in co-culture? Is this due to some metabolite cross-feeding?

Line 160: Does this mean that only 1% of the droplets were inhabited? Or in 51% and 48% of the droplets where *P. putida* and *P. veronii*, respectively, were actually present?

Line 213: This sentence is somewhat complicated to read. Consider rephrasing.

Line 434: Since the “Manual of Methods for General Bacteriology” is not openly available and difficult to obtain, a recipe of the 21C minimal media in the SI would be appreciated.

Figure specific comments:

Fig. 2E: There is a strong bimodal distribution in the data due to numerous droplets with low fluorescence at the bottom. Are these droplets with no cells or just a single cell (i.e., no real observed growth)?

Fig. 3B: Since this seems to be the only panel quantifying the time to first doubling, could the axes be zoomed in slightly more to see the data better? I understand that for other panels (cell density, growth rate) the authors decided to have uniform axes as much as possible across Figures which I appreciate.

Reviewer #3 (Remarks to the Author):

In this work titled “Fragmented micro-growth habitats present opportunities for alternative competitive outcomes”, Batsch et al explore the impact of microhabitat fragmentation on the outcome of bacterial interspecific interactions. They investigate how the division of populations into smaller, localized sub-populations across fragmented habitats, may contribute to the maintenance of diversity at the larger macro-habitat scale.

To study that, the authors contrasted growth outcomes between picoliter droplets seeded with a few founder cells per droplet and those in suspended liquid cultures. They compared the dynamics and outcomes of co-cultures across four interaction scenarios – substrate competition, substrate independence, growth inhibition, and cell killing by tailocins – using different bacterial strain combinations and media. To assess competition outcomes they used flow cytometry (cell counts), as well as time-lapse imaging and mathematical models.

The authors find that in many cases the outcome of co-cultures in picoliter droplets with low initial census population size, are different than in macro suspended liquid culture. They suggest that the observed variable and alternative interaction outcomes are a consequence of founder cell phenotypic variation and small founder population sizes. For substrate competition, this hypothesis is further supported by a mathematical model. Their main conclusion is that habitat fragmentation can lead to alternative interaction outcomes, increase variability in outcomes, and thus enhancing co-existence and diversity. The suggested underlying mechanisms is that in micro-patches seeded population are usually very small (of a few found cells) and thus one would expect phenotypic heterogeneity to

play a more important role than in well-mixed conditions at high census population size, where it would level out differences among individual founder cells.

Significance: the work is a significant and timely contribution to microbial ecology with implications extending to other fields. Microbial habitat fragmentation, an underexplored area within the field, definitely deserves more focus given its prevalence in most microbial habitats (just to name few examples: soil, plant roots and leaves, animal and human gut and skin, ocean particles). The insights provided into interspecific interactions in naturally fragmented microbial habitats are both intriguing and potentially paradigm-shifting.

Data and methodology: The droplet-in-oil technology for culturing bacteria represents a suitable and effective approach. The experimental system design, involving three bacterial pairs to represent four fundamental interaction scenarios, is elegantly executed. The reliance on flow cytometry for cell counts and different fluorescent signals to enumerate multiple species in co-cultures is methodologically sound. Time-lapse imaging of individual droplets is well designed and image processing seems to be appropriate (based on fluorescence and not single cell, see comment 5). Data presentation, including figures, is excellent and convincing. I have one comment, though, regarding the presentation of competition outcomes (see below comment 3).

Validity and robustness: The study's design and execution are robust, with appropriate biological and technical replicates and statistical analyses. The incorporation of mathematical modelling, building on previously published models (by the same authors) to include stochastic effects from phenotypic variation, enhances the study's depth.

Clarity: The manuscript is well-written offering clear and logical analysis and interpretation of complex data. However, I think that the discussion on the relative contributions of various mechanisms to the observed variability in interspecific interactions, as compared to macro-cultures, could be further clarified.

I strongly support the publication of this manuscript in Nature Communications. Although I have no significant critiques, I provide below a list of comments, questions and suggestions aimed to help improving this already impressive manuscript even more. My

hope is that these considerations will not only enhance the manuscript but also sharpen its conclusions.

1) In the substrate competition co-culture scenario involving *P. putida* and *P. veronii*, it was observed that *P. veronii*'s yield was 4 times higher compared to the macro co-culture. This was mainly attributed to the approximately 50% of droplets being solely occupied by *veronii* (solo droplets). I agree with the authors that this is likely the primary reason for the 'global' attenuation of competitive inhibition. Within the mixed droplets, *veronii* gained competitive advantage in about 24% of cases (at least based on productivity). Since mixed droplets were about 25% of all droplets (If I got it right), then it can be translated to an alternative outcome in ~5% of all populated droplets. Thus, for this interaction scenario, I think there are two key conclusions: (i) the attenuated competition inhibition was largely due to the partition of the population (with a large fraction of solo drops). (ii) in a minor portion of droplets the typically inferior competitor can win. Phenotypic variation alone can account for this alternative competition result, as demonstrated in later part of the paper and the model. I think this can be reflected better in the text and in particular in the Discussion.

Additional comments on the first interaction scenario include:

a) In the current experimental setting, there seems to be a high portion of solo droplets (~50%). As this is an important factor affecting the global outcome, it is worth noting somewhere in the text, that the frequency of solo droplets depends on the inoculum cell density, and that the number of solo droplets can vary significantly as a function of inoculum density.

b) Clarification on how yield is defined—whether as maximum (AF, OD) or at the experiment's conclusion (24 or 48 hours)—is important, especially if a decline is observed.

c) The method of calculating yield change (lines 142-3, Fig. 2A) is unclear from the provided graphs.

d) The variability in outcome in mixed droplets indicates that a predominance of mixed droplets (i.e. a situation where most droplets are mixed) could notably influence global competition. However, achieving predominantly mixed droplets would require a higher density, possibly leading to a reduction in phenotypic variation due to the increased number of founder cells in each droplet. Just some thoughts... I wonder what the authors think about it.

e) The initial cell density in the liquid suspension (5×10^5 cell /ml) was much lower than the droplets (5×10^7 cell /ml) – do you expect this to have any impact on the results that possibly has to be taken into account? Would the results and consequences be different if the suspended liquid cultured would be inoculated with the same cell density as the picodroplets? What would they predict if they reinforce small founder cells in suspended liquid macro-wells, or large initial pop in smaller droplets? I think that mentioning the difference in initial density, as well as the possible implications, worth to be mentioned in the Discussion.

f) What is expected if one inoculates a very small number of founder cells to a liquid macro culture? Do you expect to see the effect of phenotypic variation? is it really the effect of very low census seed numbers or also the limited size of the patch (i.e. the limited yield in small patches)?

g) Related to the above... is the suggested phenotypic variation in growth kinetics (growth rate, lag phase) an inherited trait? how long does this memory last? The extent to which phenotypic variation is stable or transient is of interest, particularly in response to density-dependent factors, the presence of other species or changes in media composition in co-cultures.

h) In essence, fragmentation alone – regardless to variation in interaction outcomes (or reversal of interaction) – can explain high taxonomic diversity in macro populations. This aligns with findings from Wu et al., 2022, and other, mostly macro-ecological studies, showing that spatial partitioning reduces competition and fosters diversity.

2) In both scenarios involving inhibition, only a minor fraction of droplets showed altered outcomes: 3% showed positive interactions in the first scenario, while only 0.5% of droplets in the second scenario saw tailocin-mediated removal of competitors. The ecological implications of these findings are noteworthy. I think this mostly shows that competition outcomes can be different in some small percent of patches, and this likely has ecological importance as inferior competitors can still co-exist even in very low abundance, thus maintaining diversity at the level of the macro-habitat.

3) General comment/suggestion on the presentation of competition outcomes:

I believe it worth considering another method for presenting the outcomes of species competition. Specifically, for a pair of interacting species, A and B, a graph could be designed with the initial fraction of species A on the X-axis and the fraction of species A at

the experiment's conclusion on the Y-axis. This approach offers a clearer understanding of changes in the relative abundances of competing species over time. Such a presentation would provide insights beyond mere productivity levels, revealing the competitive dynamics more accurately.

4) Mathematical model for substrate competition:

The mathematical model introduced for substrate competition, incorporating Monod growth dynamics with added stochastic elements to account for variability in growth rates and lag phases, is very elegant. Though the model looks solid and convincing, there are some differences between the observed and simulated in Fig. 7 B and C. What could explain the many more observed (experimental, Fig. 7B) droplet-outcomes with zero productivity of putida and >0 productivity of veronii compared to the simulation (Fig. 7C)?

a) Kinetic parameters in the model were taken from the suspension experiments. Why not using the mono culture in droplets parameters? Could that change anything?

b) At least from my understanding, though I might have missed something, the simulation appears not to accurately reflect the probability of solo droplets, which are a significant factor in explaining the increased productivity of veronii. If this is the case, then enhancing the model to include solo droplets occurrences could improve its alignment with empirical findings at the entire macro-habitat (including empty droplets, solo droplets and mixed ones), and offer deeper insights into the dynamics driving these competitive outcomes.

5) I assume the authors have checked it, but still mention it here to be sure. Using fluorescence (AF) rather than single-cell counts is not always a very good proxy for cell numbers, especially for small cell numbers and in cases where cells have variable metabolic/physiological states (see e.g. Ryan et al, Cell Systems 2019).

6) Some other general comments/questions/suggestions:

Line 318: The hypothesis suggesting some bacterial species might have evolved more variable growth kinetics to enhance survival under substrate competition in microhabitats presents an intriguing line of thought.

Line 323: I think that the initial paragraph of the Discussion could benefit from restructuring to emphasize more clearly the observed differences in interaction outcomes between suspended liquid cultures and fragmented habitats. Suggesting that outcomes in micro-patches can vary significantly from those inferred at the global scale due to habitat fragmentation provides a solid foundation. Following this, the discussion on the mechanisms – such as phenotypic variation and stochasticity in founder cell numbers – can be presented as likely contributors to these diverse outcomes.

Line 344: cell-cell interactions are assumed to be dominating at short (10-100 μm) distances. It would be nice to add few additional references that show this.

Lines 345-6: While phenotypic variation may occasionally allow an inferior competitor to succeed, exploring this phenomenon through a comprehensive simulation of the entire metapopulation could elucidate its broader implications on competitive dynamics and outcomes (see also comment above 4b).

Line 359: “starting cell densities (10^6 cells)”. per ml? per 140 μL ?

Lines 348-374: For improved readability, I suggest to consider dividing the long paragraph into two segments: one focusing on the overall differences in outcome, and another detailing the specific mechanisms proposed.

Lines 399-403: The role of founder cell stochasticity, including which species a cell encounters in a microhabitat and the initial number of cells of each species, could also account for the observed high diversity due to spatial fragmentation. This point warrants further exploration to differentiate its impact from that of phenotypic variation.

Line 405: I am not sure I understand what is meant in the parenthesis:” dynamic variation in spatial fragmentation”

Figures:

Fig. 2A: *P. veronii* is still growing in monoculture at 24h. This suggests that the carrying capacity has not yet been reached. Clarifying whether this aspect influences the interpretation of competitive dynamics and outcomes would be beneficial

Fig 3C: the symbols in the right are not immediately clear (mono-mixed).

Fig. S8: I am not 100% convinced that founder cell variability does not have an impact on outcome.

Minor/typos:

L. 105: missing parenthesis after “design from Duarte et al. 42”

L.191: “*veronii*, but again indifferent between mono- and co-culture conditions (Fig. 4C)” ; I think it should refer to (Fig. 3C)?

Waiving anonymity

Nadav Kashtan

REBUTTAL LETTER

We thank all three reviewers for their positive criticism and suggestions for improvement of our work. Reference to line and page numbers is with respect to the document 'Manuscript_all_changes_marked.pdf'.

Reviewer #1 (Remarks to the Author):

Summary:

I read the work of Batch and colleagues with pleasure. They use comparisons between batch and microdroplet co-culturing and combine this with theoretical modelling to study the effect of spatial fragmentation on bacterial interactions. They show that the fragmentation causes alternative outcomes than expected (based on batch culture) because of low-number fluctuations in the droplet inocula in combination with phenotypic variation within each co-cultured species.

The study is nicely quantitative, appears carefully done, and the manuscript is overall well written. The topic is timely, as understanding bacterial interactions is a key challenge to the field of bacterial communities, which play important environmental and medical roles. Their main finding is a valuable contribution worth publishing, but I have some reservations on the magnitude of the effect in natural environments.

Reply: We thank the reviewer for the appreciation of the work's implication.

Main:

1) ****My main concern is the discrepancy between the small inoculum size in this study's model and experiments (3 cells) and typical inoculum sizes in natural systems, as based on literature reports (100-1000 cells).**

The cell distribution across droplets follows a poisson distribution (LL158). Relative noise is often quantified as the standard deviation over the mean of the stochastic process. Poisson processes have a variance that is equal to the mean (λ), meaning that the relative noise scales as $1/\sqrt{\text{mean}}$. In other words, if $\lambda=3$ the relative noise is more than 50%. However, if $\lambda=100$ this decreases to 10% and with 1000 this is only 3%. That means that stochastic effects are expected to be much lower if the inoculum averages 100-1000 cells instead of 3 cells per droplet. The chance the underdog species gains stochastic numerical advantage decreases, and the effect of phenotypic variation reduces, too, because of central limit theorem. This means that fraction of droplets with an inverted outcome (compared to bulk baseline expectation) diminishes.

Reply: We thank the reviewer for this conceptual question. Indeed, one would expect a scaling effect that diminishes the propagation of stochastic sampling variation with increasing size of the founding populations. We had focused in first instance on a single droplet volume, which somewhat limits the extent of increasing founder cell populations, because larger founder cell populations will sooner run out of nutrients and go through fewer generations of cell division. To study proliferation from larger founder cell populations, one would have to increase droplet size, increasing again other variables.

*Action: We have addressed this question by two approaches. First, we conducted a new experiment with slightly larger droplet volumes (ca. 270 μ L) and two higher founder cell populations, of ca. 2-5 and 3–10 cells, for the combination of *P. putida* and *P. veronii* growing in competition on succinate. The results of this experiment show indeed that the variation in growth outcomes reduces, and the proportion of competition reversal by *P. veronii* diminishes. We present these results in a new Figure 8g-j, with corresponding new description on p. 12, lines 427-433 (results) and on p. 20 lines 837-846 (methods).*

Secondly, we conducted simulations using the droplet growth model to cover a wider range of droplet volumes and founder cell population sizes. The results from the simulations also confirm the notion that cell populations behave more homogeneously at larger founder sizes. This is now presented in the new figure 8a-f. The corresponding text was added to lines 410-426 (results).

2) Then the question is how large is a typical inoculating population? For soils, Ref. 27 is used to speculate that there are few cells per microhabitat (LL74-76), but it seems in this paper that the number of neighbours within interaction distance is about 100 cells, more in line with Ref 28 (LL341). Marine particles typically start sinking with 100-1000 founding cells (Ref 53). These numbers seem all much higher and a mismatch to the numbers used in this study. On the skin, Ref. 26 includes some modelling (their Fig. S16) that seem to suggest initial inoculation is low (e.g. 1) but that there is basically no competition in this case, as migration rates are very low. There have been some studies showing stochastically driven priority effects in for example flies (Obadia Curr Biol 2017) and worms (Vega and Gore, PLoS Biol 2017), that suggest host invasion is done by small populations (I do not know how small precisely), but this process is more complex than just fragmentation.

Reply: We thank the reviewer for bringing up this question, which indeed, is an interesting general point to think about and not exactly very well understood. However, we don't necessarily see this as a discrepancy to our study, because there is likely a range of founder cell populations relevant for particular systems and natural conditions. Colonization of the GI tract is unlikely to occur from solitary bacterial cells, but rather from thousands to millions of cells simultaneously ingested with the food. In contrast, studies of plant leaf colonization have shown that most aggregates in growth-favorable areas arise from single founder cells (10.1038/ismej.2011.209), which may be driven by the physics of microdrop formation resulting in solitary cells being enclosed (Grinberg and Kashtan, 2021). Confocal scanning images of plant roots grown in soils inoculated with fluorescently tagged bacteria also show both solitary cells as well as small aggregates (Tovi et al., 2021). The reference to the soil estimates (100-cell aggregates at ca $1e9$ cells/gram soil), is a static estimate and does not inform on founder cell numbers (if 100 cells is a steady-state estimate, it is rather unlikely that the founder population would be similar since that would obviate intrinsic growth). Finally, it also seems unlikely that marine particles start with founder cell populations of 100-1000 cells – maybe this is the point where they start sinking and are observed, but not necessarily the real founder size. But indeed, this leads to the more general question of the effect of founder cell population sizes on the variability of growth and interaction outcomes. This is a topic that has recently been studied in the context of lag phase variation in reference 10.1093/femsml/uqac022.

Action: See point 1 above on the new wetlab experiments and model simulations.

We further reorganized the discussion based on comments from all reviewers, to better reformulate the question of founder cell size and habitat colonization in more general terms. Paragraph on p. 14, l. 504-526 discusses phenotypic variation and the paragraph starting on p. 16 l. 547-563 focuses on the question of starting populations in pristine habitats.

3) The authors should at least comment on this discrepancy. It seems easy to do some simulations to assess the magnitude of the effect with larger founding populations, and perhaps some additional experiments can be done with larger droplets and more cells per droplet.

Reply and action; see our replies above to points 1 and 2.

4) ******Then the comparison between batch and microdroplets appears somewhat convolved. This is because placing cells in microdroplets changes multiple things at once: effects of fragmentation in the sense of compartmentalisation (e.g spatial isolation) and effects of densities/spatial structuring (e.g average cell-cell distances).

Reply: We thank the reviewer for this comment. See our detailed replies below to point 5 and the new text in the discussion on p.14, l. 504-510.

5) As for the compartmentalization, the small volume means that only a modest population of 100 cells per droplet gives a bulk-equivalent concentration of $1E10$ cells/mL, meaning populations in droplets may reach steady state much faster (in approx 5 generations). It is therefore not clear if batch culture and droplets are in the same state after 24 hours. Or could the large variation even among solo droplets indicate the cultures have not reached a steady-state?

Reply: We thank the reviewer for these considerations. We started off with the same substrate concentration for picoliter droplet and for the bulk liquid suspension cultures. Therefore, there should not be an a priori difference in the extent of growth or the attained density in either setup. We also measured the extent of growth in the droplets by collapsing the droplet emulsion and quantifying the cell numbers in solution, which is the same as in bulk suspension (Fig. 2C and D). Batch culture and droplets should thus be in the same state after 24 h (although it is possible that some of the productivity variation in solo droplets originates from individual populations having started slightly later and not having reached stationary phase at the 24 h sampling point (compare, e.g., Fig 2E and Fig 6C with the time-series).

Action: We have included this argument in the text on p. 14 l., 482-488.

6) For the spatial organisation: 35 μ L corresponds to a droplet diameter of 31 μ m, suggesting typical cell cell distances of about ~ 15 μ m. At 10^5 cells/mL, the founding concentration in batch culture, this is much larger, around 200 μ m. To what extent does this distance change the interaction magnitudes? Apart from the killing story, which has an obvious distance effect, the authors do not really discuss the spatial aspect, while this actually generates more behavioral degrees of freedom in case of co-cultures. For example, a recent study has shown that species can change the spatial organisation of the interaction partner (e.g. D'Souza et al, PNAS 2023). I think this point can be

addressed with mostly textual changes, by clearly distinguishing the two effects microdroplet compartmentalization causes, and clarifying which aspect the authors refer to in each case.

Reply: Droplets with a diameter of 40 μm (as we produce) would have a volume of 33.5 pL ($\frac{4}{3}\pi r^3$). Both droplet and bulk liquid suspended cultures are started with the same suspension, resulting in the same starting densities. On average, we start with a suspension of 1.8×10^7 cells. In bulk liquid suspension, this would give an average individual culture volume per cell of 55 pL and cell-to-cell distance of 24 μm .

However, indeed, because the actual starting number in droplets is a Poisson process, droplets with two cells would have a slightly higher density and only 16 μm cell-cell distance. The starting distribution of cells per droplet is presented in Figure S2, to illustrate this.

The reviewer is also right that cells in droplets would be restricted to the droplet boundaries, whereas cells in the bulk suspension could potentially also interact with cells much further away, since their movement is not restricted. But this is part of the mechanism that we are studying here and an effect of the fragmentation.

Action: We clarified the starting densities in the Material section in l. 799-802, and specified the cell-cell-distances in the text in l. 540-542. We discussed how fragmentation confines cells to the same co-localized volume and therefore enforces interactions in a localized area between cells with specific (phenotypic) variations, whereas bulk conditions may allow the variation in cell-partner over time, therefore limiting the local impact of individual cell variabilities. See lines 605-611.

7) *The paper is overall well written but could be improved with some streamlining. The killing story seems somewhat like a separate story within this paper. The authors could move some material to the supplement to focus on the key ideas in the main text and figures.

Reply and action: We thank the reviewer for the suggestion. However, we would prefer to keep the tailocin story as part of the paper, since it demonstrates that the principle does not only hold for metabolic interactions. We have restructured the discussion for clarity on the processes underlying variable growth outcomes in droplets.

Minor:

8) *The authors focus mostly on phenotypic variation of growth kinetic parameters, e.g the kind that is expected from stochastic gene expression (see Ackermann, Nat Rev Microb 2015 for a review). This is certainly interesting, but the population-level effects of such variation quickly averages out in populations larger than a 1000 cells (see above). Instead, I expect potentially much stronger effects of phenotypic variation that is bistable and heritable (for example the glycolysis-glycogenesis switch, Basan et al Nature 2020, or cell states such as bacterial dormancy), as these tend to preserve the variation to much higher cell numbers. Experiments with conditions and strains that show such large variations could be interesting to explore in future work, but perhaps the authors can already comment such scenarios.

Reply: We thank the reviewer for bringing up this point. Indeed, we don't specifically suggest that phenotypic variation in our cases is due to differences from stochastic gene expression. However, tailocin expression in P. protegens is considered a case of stochastic gene expression variation (10.1038/s42003-020-01581-1). We do not know the nature of the phenotypic variation among cells of e.g., P. putida or P. veronii, which may be similar as described for e.g., lag time variation in P. fluorescens (10.1093/femsml/uqac022).

Action: we have revised the text on p. 14/15 to be more specific about phenotypic variation (l. 509-526) and have added several new references about this aspect.

9) L66: Ref.30 does show spatial organisation of the microbial community in the mouse gut, but I do not think from that can be concluded that there are microhabitats as this requires isolation.

Reply: We appreciate this comment, but would like to mention that particle growth also can be regarded as 'isolated' growth, since the cells will remain within their microcolonies or biofilms, and only later would 'slough off' to seed to habitats. For a recent illustration of this, see BioRxiv d'Souza manuscript 2023.07.14.548877v2.full. Also, the paper by Bergstrom (10.1126/science.aay7367) on fecal pellet formation by mucus in the mouse gut is suggestive for compartments being formed that shield bacterial groups.

Action: We modified this text (l. 80) and added the Bergstrom reference as an illustration of encapsulation of microbial cells in the gut. See further the changes in the discussion (l. 547-561) where we discuss founder cell number effects.

10) L318: "This may also imply that some bacterial species have become selected for more variable growth kinetics ". It's probably true that the mean value as well as the variation of growth rate are both subject to natural selection, but it is unclear why bacteria would involve a large variation in growth rate if they have the ability to just evolve a higher average growth rate.

Reply: We thank the reviewer for raising this interesting thought. However, there is no a priori reason why both cases could not exist. An average high growth rate may bring additional disadvantages, such as the need for a constant high supply of food and corresponding proteome allocation. A stochastic phenotypic high growth rate may give temporary advantage, while maintaining flexibility. Modeling of E. coli lag phase under antibiotic tolerance by Moreno-Gamez (10.1073/pnas.2003331117) indeed suggests evolutionary advantages for extreme lag time variation.

Action: We modified the text in the discussion on p. 17, l. 606-611 to include selection on physiological variation.

11) Perhaps the authors can include infection biology in their list of examples with compartmentalization (as reviewed by for example Azimi et al, Nat Rev Microb 2022)

Reply and action: We thank the reviewer for the suggestion, but have refrained from adding this reference, because it would require discussing dispersal mechanisms more in general terms – which

we feel distract from the focus of the study.

12) I think a very nice documented example of compartmentilization is the encapsulation by mucus in the mouse gut (Bergstrom, Science 2020). But I dont know how large the population sizes are in that case.

Reply and action: We thank the reviewer for mentioning this paper. Unfortunately, the results in this paper do not allow to draw conclusions on the volume of the fecal pellets and the resulting microbiota-size inside. Sections only report the density of bacteria from the distance to the mucus layer – but to extrapolate this to 3D is too complicated. We cite this reference in the context of line 80.

13) I think it would be nice to have cell number instead of the fluorescence signal when quantifying bacterial abundance of the two strains in each particle, I think the authors have the data to calibrate this (e.g. fluorescence per cell).

Reply: We thank the reviewer for this suggestion. We agree that cell counts would have perhaps been more illustrative for population growth than biomass-inferred by fluorescence. However, the fluorescence per-cell changes over time (see the plot below, which shows the fluorescence normalized by OD over time of monoculture growth), there is cell movement in individual droplets (within the exposure time), some cells in droplet are out-of-focus and cells can clump, which make it difficult to accurately estimate cell numbers over time. Taking the fluorescence area AND the fluorescence intensity is the best compromise here. Single droplet biomass growth curves illustrate this nicely, we believe (Fig. 6).

To accurately quantify total cell numbers, we coalesce all droplets to a single emulsion and count cells by flow cytometry, as reported.

Action: We clarified the use of area x fluorescence (AF) instead of approximating cell-numbers on p. 22 in lines 910-924.

14) This is optional, but I was wondering if the authors could say anything on the current debate on the prevalence of positive vs negative interactions in microbial communities. Does fragmentation make competition or cooperation more likely?

Reply and action: We thank the reviewer for bringing up this point and refer to reference Wu et al (2022) who developed fragmentation models under mostly positive or negative interactions (but did

not specifically consider phenotypic variation). To focus the discussion in the current text on the dominance of positive or negative interactions would be beyond what we can reasonably extrapolate from our experiments. Therefore, we keep the discussion on the effects of spatial fragmentation on phenotypic variation in more general terms. See the last paragraph of the discussion (l. 606-615).

Reviewer #2 (Remarks to the Author):

The work presented by the authors is a compelling and timely study of how phenotypic variation can determine the outcome of different microbial species interactions due to isolation in microhabitats. They use droplet microfluidics combined with classical microbial culturing approaches to tease apart the influence of standing phenotypic variation versus the “expected” outcome based on traits quantified from bulk culturing experiments such as the growth rate, lag-time, and yield. The authors test four scenarios: direct competition, indifference, inhibition, and direct cell killing. In a significant fraction of the microdroplet incubators, the expected outcome of the final population composition is reversed. The authors further employ a stochastic model of population co-growth for the competition scenario fed with parameters inferred from the experimental results to test the relative importance of traits versus the stochastic phenotypic variation in the founder population. The authors conclude that this variation is a key driver of alternative interaction trajectories and can contribute to the maintenance of diversity.

This is a well-written potentially impactful manuscript that may benefit from some clarification and additional experiments/ simulations to disentangle the contribution of different mechanisms. Below, I outline some major comments followed by minor comments and figure-specific points. Overall, I am excited about this study which highlights the importance of studying microbial interactions and ecological principles at the scale that matters to individual cells in microhabitats.

Reply: We thank the reviewer for the positive comments on our work.

Major comments:

1) The authors use combinations of different species and substrates to tease out the resulting population dynamics depending on competition, indifference, or antagonism (inhibition or killing). While this can highlight the importance of the standing phenotypic variation, it is confounded by the species-specific trait variations. As an interesting control, I would suggest using a single species (*P. putida* or *P. veronii*) tagged with two different fluorescent proteins and performing the same experiment in co-culture and fragmented growth. This would also provide excellent data to validate model predictions.

*Reply: We thank the reviewer for the suggestion. We have repeated experiments with a coculture of isogenic *P. putida* except for the fluorescence gene insertion. This shows similar results as for both *P. protegens* isolates (but without the tailocin effect, Fig. 5) – that there is an equilibrated distribution of biomass of either species across droplets (of note that there is a slight effect of the reporter protein production on strain fitness).*

Action: This is now included in Supplementary Figure 5, with explanation in l. 749 (methods), Supplementary Table 4 (experimental conditions), l. 207-213 (results).

2) Although there is a section in the discussion devoted to the expected cell cluster sizes in e.g., soil and marine environments, the overall size of the droplets and cell numbers employed here are lower than what may be expected in natural environments. I believe there is an opportunity to use this experimental system (by increasing the droplet size or founder population) or the mathematical model (tuning the initial population size) to tease out if there is a threshold or simply a gradual change where species-specific kinetic parameters (the mean) becomes more important than the phenotypic heterogeneity in the founder population. Such an analysis will provide an important contribution to understanding both the importance of fragmentation in natural environments and also guide future experimental and engineering efforts when using droplet microfluidics. In other words, can the authors derive a critical cluster size upon which selection overrides the influence of the initial standing variation?

Reply: We thank the reviewer for bringing this up and we refer further to our reply to both other Reviewers on the same points. We have addressed this both by extending model simulations and by running additional experiments with two different higher founder cell population sizes in slightly larger droplets. This suggests that the phenotypic variation effect levels off above 10 cells, although it does not disappear.

Action: We have addressed this question by two approaches. First, we conducted a new experiment with slightly larger droplet volumes (ca. 270 pL) and two higher founder cell populations, of ca. 2-5 and 3-10 cells, for the combination of P. putida and P. veronii growing in competition on succinate. The results of this experiment show indeed that the variation in growth outcomes reduces, and the proportion of competition reversal by P. veronii diminishes. We present these results in a new Figure 8g-j, with corresponding new description on p. 12, lines 427-433 (results) and on p. 20 lines 837-846 (methods).

Secondly, we conducted simulations using the droplet growth model to cover a wider range of droplet volumes and founder cell population sizes. The results from the simulations also confirm the notion that cell populations behave more homogeneously at larger founder sizes. This is now presented in the new figure 8a-f. The corresponding text was added to lines 410-426 (results).

3) When assessing the growth trajectories in individual droplets, a consistent pattern of rapid P. putida and lesser P. veronii productivity is observed, followed by a decline in P. putida productivity but a further increase in P. veronii productivity (Fig. 6E and Fig. S8). Depending on the recipe of the 21C minimal media (which I could not locate in the book reference of Gerhard et al. 1984 and only a PDF advert for the book is provided by PubMed when searching for the reference), P. veronii (and S. wittichii) may use alternative electron acceptors if the droplet experiences anoxia following rapid consumption by the growing population. Although this most likely does not influence the maximum observed growth rate, it can shift the balance in the final observed biomass within the droplets after 24 to 48h (especially at the high cell densities observed by the authors as shown in Fig. 2D).

Reply: We thank the reviewer for the comment and apologize that we had not realized that Gerhard's reference book is not available online.

*Indeed, *P. veronii* is able to denitrify and to (likely) grow better under microaerophilic conditions than *P. putida* (see e.g., Borer B, Tecon R, Or D. 2018. Nat Commun 9:769. <https://doi.org/10.1038/s41467-018-03187-y>). However, the 21C medium does not contain nitrate (only ammonium); therefore, nitrate respiration is unlikely. We cannot exclude other electron acceptors or utilization of metabolites arising from the partner's metabolism at later stages of growth. However, in plate reader experiments we also see an increase of fluorescence in stationary phase but no increase of culture turbidity (see the plot in response to Reviewer 1 under point 13). Therefore, we think the most likely explanation for this increase is continued expression from the gene reporter.*

Action: We provide the recipe of 21C in the Supplementary Table 3. We explained the use of Area-x-Fluorescence for biomass estimation and its possible pitfalls more in detail in l. 910-914.

4) One difficulty when assessing the results is the use of normalized fluorescence intensity to discern different conditions inside droplets. More specifically, I cannot understand from this data (e.g., Fig. 3D) if the droplets where *P. veronii* dominates are due to an initial droplet that did have both species but only *P. veronii* was able to grow (and *P. putida* was semi-dormant), or if *P. veronii* outcompetes the latter. A comparison of the normalized fluorescence of the founding community (media lacking an essential nutrient when creating the droplets or even including a growth inhibitor/antibiotic) and immediate quantification of the fluorescence could be an avenue to distinguish these cases (also for the bimodal distribution of normalized productivity in Fig. 2E, 4D, 5B, and 7B). Alternatively, since I believe images were taken at timepoint 0, these could also be used directly to establish a baseline fluorescence signal.

Reply: We thank the reviewer for this comment. Since these are time point samples, we have no a priori way to place a threshold below which we consider a cell non-growing or dormant – in contrast to the time-lapse droplet image data, where populations can be followed in individual droplets over time. For example, in Figure 6c one can see some cell fluorescence traces very close to 0, suggesting that these don't grow. In comparison with T=0 plots there is a certain proportion of cells where one could assume that either one or the other, or both founder cells have not further divided (e.g., the plot fractions a, c and d in Figure 4d).

*Action: To show this more clearly, we have added a Supplementary figure 4 with the T0 and T24 comparisons. The text on p. 22, l. 920-924 was modified, and the results description in l. 207 was changed to 'was dominated by *P. veronii*'. The issue of dormant cells was discussed in the paragraph between l. 521-526, and the effect of potentially dormant cells in our results was further mentioned in l. 207-213, l. 261-262 and l. 521.*

Minor comments (mostly details and phrasing):

5) Line 21: In my view it is more the "effect of being constrained within a microhabitat on the emergence of bacterial interspecific interactions" instead of the effect of a microhabitat.

Reply: we thank the reviewer for the suggestion and have modified the abstract accordingly.

6) Line 38: Some jargon is used without further explanation in the introduction which may diminish the readability, especially for a journal with a broad scope such as Nature Communication. Brief definitions for concepts common in microbial ecology such as founder cells or standing phenotypic variation would be appreciated.

Reply: Point well taken. See lines 102-104 for explanation of founder cells. 'Standing' was removed here.

7) Line 86: This sentence is difficult to understand. I would consider rephrasing to ensure that the meaning comes across as this is a critical sentence of the paper. I am not sure that microhabitats themselves favor the standing variation (or anything as a matter of fact, maybe promote? Or facilitate?).

Reply: Was corrected to 'Our hypothesis here was thus that microhabitats would enable existing phenotypic variation among individual founder cells (i.e., the cells present in a habitat that give rise to new offspring), to have stronger effects on reproductive success'.

8) Line 105: Closing bracket missing after the sentence.

Reply: Thank you. Was corrected.

9) Line 136: Is there a hypothesis of why *P. veronii* grew slightly faster in co-culture? Is this due to some metabolite cross-feeding?

Reply: We thank the reviewer for the observation. As we speculate in PLoS Comput Biol 19:e1011402, this may be due to metabolite cross-feeding. This was added here (l. 164-165).

10) Line 160: Does this mean that only 1% of the droplets were inhabited? Or in 51% and 48% of the droplets where *P. putida* and *P. veronii*, respectively, were actually present?

*Reply: We apologize. This was awkwardly formulated. We corrected this to: 'Imaging after 24 h indicated that ca. 26 % and 23 % of droplets from co-cultures were occupied solely by either *P. putida* or *P. veronii*, respectively (solo droplets), and with 26 % containing both (mix droplets; the other 25% being empty).' (lines 194-196)*

11) Line 213: This sentence is somewhat complicated to read. Consider rephrasing.

Reply: Thank you. Was rephrased to 'To explore whether fragmented growth in other situations than substrate competition would also result in more variable outcomes than in mixed liquid culture, we used two further strain combinations, which illustrate an inhibition and a killing interaction' (l. 280-282)

12) Line 434: Since the “Manual of Methods for General Bacteriology” is not openly available and difficult to obtain, a recipe of the 21C minimal media in the SI would be appreciated.

Reply: We hadn't realized this and apologize. We have included the 21C recipe in a new supplementary table 4.

Figure specific comments:

13) Fig. 2E: There is a strong bimodal distribution in the data due to numerous droplets with low fluorescence at the bottom. Are these droplets with no cells or just a single cell (i.e., no real observed growth)?

Reply: These are droplets that have detectable fluorescence of the 'other' channel. Therefore, we assume they have a single non-growing cell. See our remarks above to point 4 and the new Supplementary figure 4 plus corresponding descriptions.

14) Fig. 3B: Since this seems to be the only panel quantifying the time to first doubling, could the axes be zoomed in slightly more to see the data better? I understand that for other panels (cell density, growth rate) the authors decided to have uniform axes as much as possible across Figures which I appreciate.

Reply: Axis were revised to zoom in more (scale 0-12 instead of 0-20). We prefer to keep the same scale for both to avoid explanation that there is a different scale.

Reviewer #3 (Remarks to the Author):

In this work titled “Fragmented micro-growth habitats present opportunities for alternative competitive outcomes”, Batsch et al explore the impact of microhabitat fragmentation on the outcome of bacterial interspecific interactions. They investigate how the division of populations into smaller, localized sub-populations across fragmented habitats, may contribute to the maintenance of diversity at the larger macro-habitat scale.

To study that, the authors contrasted growth outcomes between picoliter droplets seeded with a few founder cells per droplet and those in suspended liquid cultures. They compared the dynamics and outcomes of co-cultures across four interaction scenarios – substrate competition, substrate independence, growth inhibition, and cell killing by tailocins – using different bacterial strain combinations and media. To assess competition outcomes they used flow cytometry (cell counts), as well as time-lapse imaging and mathematical models.

The authors find that in many cases the outcome of co-cultures in picoliter droplets with low initial census population size, are different than in macro suspended liquid culture. They suggest that the observed variable and alternative interaction outcomes are a consequence of founder cell phenotypic variation and small founder population sizes. For substrate competition, this hypothesis is further supported by a mathematical model. Their main conclusion is that habitat fragmentation can lead to alternative interaction outcomes, increase variability in outcomes, and thus enhancing

co-existence and diversity. The suggested underlying mechanisms is that in micro-patches seeded population are usually very small (of a few found cells) and thus one would expect phenotypic heterogeneity to play a more important role than in well-mixed conditions at high census population size, where it would level out differences among individual founder cells.

Significance: the work is a significant and timely contribution to microbial ecology with implications extending to other fields. Microbial habitat fragmentation, an underexplored area within the field, definitely deserves more focus given its prevalence in most microbial habitats (just to name few examples: soil, plant roots and leaves, animal and human gut and skin, ocean particles). The insights provided into interspecific interactions in naturally fragmented microbial habitats are both intriguing and potentially paradigm-shifting.

Data and methodology: The droplet-in-oil technology for culturing bacteria represents a suitable and effective approach. The experimental system design, involving three bacterial pairs to represent four fundamental interaction scenarios, is elegantly executed. The reliance on flow cytometry for cell counts and different fluorescent signals to enumerate multiple species in co-cultures is methodologically sound. Time-lapse imaging of individual droplets is well designed and image processing seems to be appropriate (based on fluorescence and not single cell, see comment 5). Data presentation, including figures, is excellent and convincing. I have one comment, though, regarding the presentation of competition outcomes (see below comment 3).

Validity and robustness: The study's design and execution are robust, with appropriate biological and technical replicates and statistical analyses. The incorporation of mathematical modelling, building on previously published models (by the same authors) to include stochastic effects from phenotypic variation, enhances the study's depth.

Clarity: The manuscript is well-written offering clear and logical analysis and interpretation of complex data. However, I think that the discussion on the relative contributions of various mechanisms to the observed variability in interspecific interactions, as compared to macro-cultures, could be further clarified.

I strongly support the publication of this manuscript in Nature Communications. Although I have no significant critiques, I provide below a list of comments, questions and suggestions aimed to help improving this already impressive manuscript even more. My hope is that these considerations will not only enhance the manuscript but also sharpen its conclusions.

Reply: We thank the reviewer for the positive criticisms and excellent suggestions.

1) In the substrate competition co-culture scenario involving *P. putida* and *P. veronii*, it was observed that *P. veronii*'s yield was 4 times higher compared to the macro co-culture. This was mainly attributed to the approximately 50% of droplets being solely occupied by *veronii* (solo droplets). I agree with the authors that this is likely the primary reason for the 'global' attenuation of competitive inhibition. Within the mixed droplets, *veronii* gained competitive advantage in about 24% of cases (at least based on productivity). Since mixed droplets were about 25% of all droplets (If I got it right), then it can be translated to an alternative outcome in ~5% of all populated droplets.

Thus, for this interaction scenario, I think there are two key conclusions: (i) the attenuated competition inhibition was largely due to the partition of the population (with a large fraction of solo drops). (ii) in a minor portion of droplets the typically inferior competitor can win. Phenotypic variation alone can account for this alternative competition result, as demonstrated in later part of the paper and the model. I think this can be reflected better in the text and in particular in the Discussion.

Reply: This is correct. We have explained this better in the text and discussion paragraph between l. 572-594.

Additional comments on the first interaction scenario include:

a) In the current experimental setting, there seems to be a high portion of solo droplets (~50%). As this is an important factor affecting the global outcome, it is worth noting somewhere in the text, that the frequency of solo droplets depends on the inoculum cell density, and that the number of solo droplets can vary significantly as a function of inoculum density.

Reply: This is correct. It is a consequence of the Poisson nature of the droplet formation and the cell density in the suspension that is used to produce the droplets (see, for example, the initial cell counts in droplets in mono-culture of Supplementary figure 2).

Action: we clarified this by changing the text in lines 572-594.

b) Clarification on how yield is defined—whether as maximum (AF, OD) or at the experiment's conclusion (24 or 48 hours)—is important, especially if a decline is observed.

Reply: We use two yield definitions here. First, the direct cell counts by flow cytometry. Secondly, the area of the segmented cells times their fluorescence. We do this last part to correct for cell clumps, out-of-focus cells and moving cells inside droplets. Since we don't use scanning microscopy here, we compensate for thicker clumps by the on average higher fluorescence.

Action: This is explained on p. 22 in the paragraph between lines 910-914.

c) The method of calculating yield change (lines 142-3, Fig. 2A) is unclear from the provided graphs.

Reply: Figure 2A is used to evaluate growth rates as explained in lines 160-162; not necessarily growth yields, because of the difficulty to convert fluorescence to cell biomass. See Figure S3 for the equivalent culture turbidity measurements.

Action: We hope the method of calculating yield change is sufficiently explained in the revised Methods section (p. 22, l. 910-918).

d) The variability in outcome in mixed droplets indicates that a predominance of mixed droplets (i.e. a situation where most droplets are mixed) could notably influence global competition. However, achieving predominantly mixed droplets would require a higher density, possibly leading to a reduction in phenotypic variation due to the increased number of founder cells in each droplet. Just some thoughts... I wonder what the authors think about it.

Reply: See our replies to Reviewers 1 and 2 about their question on larger founder populations. We believe one can argue as to whether 'natural' situations favor very small or larger founder cell populations. What we show here is that if founder cell populations are low (few cells per habitat), phenotypic variation effects become more important, as opposed to larger founder cell populations.

*Action: We have conducted additional simulations and experiments with *P. putida* and *P. veronii* in slightly bigger droplets, with two larger founder populations (ca. 3–5 and 5–10 cells per species per droplet) to demonstrate that effects of individual variations decrease with increasing founder population sizes. This is now shown in a new Figure 8 and the additional Supplementary figures 12–14. See further the revised discussion.*

e) The initial cell density in the liquid suspension (5×10^5 cell /ml) was much lower than the droplets (5×10^7 cell /ml) – do you expect this to have any impact on the results that possibly has to be taken into account? Would the results and consequences be different if the suspended liquid cultured would be inoculated with the same cell density as the picodroplets? What would they predict if they reinforce small founder cells in suspended liquid macro-wells, or large initial pop in smaller droplets? I think that mentioning the difference in initial density, as well as the possible implications, worth to be mentioned in the Discussion.

Reply: We thank the reviewer for this observation and we apologize for this confusion (see also our replies to point 6 of Reviewer 1). Both bulk liquid and droplet growth were started with the same suspension (one part transferred to a 96-well plate, the other to the microfluidics droplet generator). However, it is correct that actual droplets that happen to have 2 founder cells start with a higher density than the average bulk suspension (as explained above). This is an inevitable effect of the Poisson distribution in the encapsulation process.

Since the culture medium has the same nutrient content, the nutrient availability is the same for cells in both situations, and the final cell density is the same (again, what we roughly show with flow cytometry measurements on collapsed droplets in Fig. 2d).

Action: We specified this in l. 171/172 of the first-time results, l. 482-487 of the discussion, and l. 799-802 of the Methods section.

f) What is expected if one inoculates a very small number of founder cells to a liquid macro culture? Do you expect to see the effect of phenotypic variation? is it really the effect of very low census seed numbers or also the limited size of the patch (i.e. the limited yield in small patches)?

Reply: This is an interesting question, because we really don't know whether standing phenotypic variation in founder cells is propagated exactly to their progeny, and, if so, for how many generations this would be propagated (some suggestions on this can be found in reference Microlife 3, uqac022 (2022)). We can only conclude that in the comparison of droplet (pL) and regular liquid culture volumes (140 μ l), and allowing the same number of generations of population growth, the droplets cause more variation in growth outcomes.

Action: No further action.

g) Related to the above... is the suggested phenotypic variation in growth kinetics (growth rate, lag phase) an inherited trait? how long does this memory last? The extent to which phenotypic variation is stable or transient is of interest, particularly in response to density-dependent factors, the presence of other species or changes in media composition in co-cultures.

Reply: This is an interesting question, indeed, which we cannot solely address on the basis of results here. In our related work with surface-grown microcolonies (BioRxiv 10.1101/2024.05.19.594856) we confirm highly variable growth rates from single cells – which seem to be propagated at the level of microcolonies (<50 cells). Since this is not peer-reviewed yet, we do not wish to cite this here.

Action: We have restructured the discussion to include a paragraph on phenotypic variation more specifically (l. 504-526).

h) In essence, fragmentation alone – regardless to variation in interaction outcomes (or reversal of interaction) – can explain high taxonomic diversity in macro populations. This aligns with findings from Wu et al., 2022, and other, mostly macro-ecological studies, showing that spatial partitioning reduces competition and fosters diversity.

Reply and action: Thank you. We already cited the Wu et al., 2022 paper in the introduction and have discussed the more general implications on diversity in the last paragraph of the discussion (also modified and cited macro-ecological work). Of note that habitat fragmentation in macro-ecology also has a negative connotation to refer to habitat disruption as a result of human activities, leading to isolated populations and less gene mixing. See Lines 596-601.

2) In both scenarios involving inhibition, only a minor fraction of droplets showed altered outcomes: 3% showed positive interactions in the first scenario, while only 0.5% of droplets in the second scenario saw tailocin-mediated removal of competitors. The ecological implications of these findings are noteworthy. I think this mostly shows that competition outcomes can be different in some small percent of patches, and this likely has ecological importance as inferior competitors can still co-exist even in very low abundance, thus maintaining diversity at the level of the macro-habitat.

Reply and action: We thank the reviewer for the comment. Also on the basis of further comments by the same reviewer we have largely restructured the discussion.

3) General comment/suggestion on the presentation of competition outcomes:

I believe it worth considering another method for presenting the outcomes of species competition. Specifically, for a pair of interacting species, A and B, a graph could be designed with the initial fraction of species A on the X-axis and the fraction of species A at the experiment's conclusion on the Y-axis. This approach offers a clearer understanding of changes in the relative abundances of competing species over time. Such a presentation would provide insights beyond mere productivity levels, revealing the competitive dynamics more accurately.

Reply and action: We thank the reviewer for the suggestion. Unfortunately, we did not have more time samples available. As a compromise, and in response to point 4 from Reviewer 2, we plot the T0

and T24 or T48 data in a comparative manner (Supplementary figure 4), and add corresponding results/discussion. The true competitive manner can be viewed from Figure 6 with the time-lapse imaging.

4) Mathematical model for substrate competition:

The mathematical model introduced for substrate competition, incorporating Monod growth dynamics with added stochastic elements to account for variability in growth rates and lag phases, is very elegant. Though the model looks solid and convincing, there are some differences between the observed and simulated in Fig. 7 B and C. What could explain the many more observed (experimental, Fig. 7B) droplet-outcomes with zero productivity of putida and >0 productivity of veronii compared to the simulation (Fig. 7C)?

Reply: We thank the reviewer for bringing up these points and apologize for the lack of clarity here. Our goal was not to reproduce the empirical data by the model (which we could have done by fitting), but to predict the effects of underlying variables.

There are probably two reasons for the differences between empirical data and model outcomes: first, we set a 15% probability for founder cells to remain non-growing at the beginning of each simulation. This is based on the average proportion of droplets with low AF-values that we observed for PPU and PVE at T24 and T48, compared to T0 (see Supplementary Fig. 4). The second reason is that we used a slightly higher average cell number in the simulations (3 cells per species) than what we measured in the droplets (~1-2 cells per species in mix-droplets, see Fig. S2-C), which thus decreases the probability for a growth-impaired cell to be alone. We wanted to favour the occurrence of mix droplets in the simulations and focus on droplets with about ~2-3 cells as founding population sizes per species.

a) Kinetic parameters in the model were taken from the suspension experiments. Why not using the mono culture in droplets parameters? Could that change anything?

Reply: Because the parameters from suspension experiments come from large populations of cells, they better reflect the average growth kinetics for either of the species in the used growth medium. This then allowed to test the effects of growth kinetic variability (e.g., as in Fig. 7g-i). Using droplets to infer these parameters would also be interesting, but we felt we had too few mono-culture droplets in the timelapse experiment to deduce this properly. Additionally, the growth kinetics of P. veronii and P. putida seems slightly different in the timelapse experiments compared to the timepoint experiment, due to the oxygen level being altered (as mentioned in line 875). Thus, we preferred to not extrapolate any mean parameters from these.

Action: No further action.

b) At least from my understanding, though I might have missed something, the simulation appears not to accurately reflect the probability of solo droplets, which are a significant factor in explaining the increased productivity of veronii. If this is the case, then enhancing the model to include solo droplets occurrences could improve its alignment with empirical findings at the entire macro-habitat (including empty droplets, solo droplets and mixed ones), and offer deeper insights into the dynamics driving these competitive outcomes.

Reply: Indeed, we did not take solo droplets into account in the simulations (cell numbers between 1–3), in order to be able focus on the effects of phenotypic variability and stochastic starting cell number differences (and not necessarily on bulk outcomes).

*Action: We included new simulations with the effect of increased starting population numbers (Fig. 8), for which we added a bulk-scale expected outcome of *P. veronii*-*P. putida* ratios, both in absence or presence of solo droplets (Supplementary figure 12). See l. 420-422.*

5) I assume the authors have checked it, but still mention it here to be sure. Using fluorescence (AF) rather than single-cell counts is not always a very good proxy for cell numbers, especially for small cell numbers and in cases where cells have variable metabolic/physiological states (see e.g. Ryan et al, Cell Systems 2019).

Reply: We thank the reviewer for the comment. Fluorescence per cell can indeed vary (as shown in the paper of Ryan et al), and this is clearly visible from our timelapse images. We refer to our response to reviewers 1&2 about the question of AF use instead of cell count. We also add that we decided to rather keep the fluorescence area x the mean fluorescence, as a better way to account for the large clump of cells where several cells can overlap on the same position of the image, with the idea that the higher concentration of cells also leads to higher fluorescence on the picture.

Action: We clarified why we use AF measures instead of approximating cell numbers in lines 910-916.

6) Some other general comments/questions/suggestions:

Line 318: The hypothesis suggesting some bacterial species might have evolved more variable growth kinetics to enhance survival under substrate competition in microhabitats presents an intriguing line of thought.

Reply and action: Thank you for the comment. You can also see above our reply to Reviewer 1 about this hypothesis. See the revised discussion on p.17, l. 605-611.

7) Line 323: I think that the initial paragraph of the Discussion could benefit from restructuring to emphasize more clearly the observed differences in interaction outcomes between suspended liquid cultures and fragmented habitats. Suggesting that outcomes in micro-patches can vary significantly from those inferred at the global scale due to habitat fragmentation provides a solid foundation. Following this, the discussion on the mechanisms – such as phenotypic variation and stochasticity in founder cell numbers – can be presented as likely contributors to these diverse outcomes

Reply and action: We appreciate the suggestion and have restructured the discussion to better highlight the different processes that may underlay the observed growth outcome variations in droplets. First paragraph of the discussion refocused (p. 13, l. 447-460).

8) Line 344: cell-cell interactions are assumed to be dominating at short (10-100 μm) distances. It would be nice to add few additional references that show this.

Reply and action: Discussion paragraph on relevance of the used droplet volumes revised on p. 15, l.528-542.

9) Lines 345-6: While phenotypic variation may occasionally allow an inferior competitor to succeed, exploring this phenomenon through a comprehensive simulation of the entire metapopulation could elucidate its broader implications on competitive dynamics and outcomes (see also comment above 4b).

Reply: Thank you again for the suggestion. See our response to point 4b, and further the revised paragraph on p. 16/17, l. 547-563.

10) Line 359: “starting cell densities (10^6 cells)”. per ml? per 140 μ L?

Reply and action: 10^6 cells per 140 μ L. Thank you for the note. We clarified this in line 463.

11) Lines 348-374: For improved readability, I suggest to consider dividing the long paragraph into two segments: one focusing on the overall differences in outcome, and another detailing the specific mechanisms proposed.

Reply and action: Thank you, we adjusted as suggested. Restructuring and a new paragraph on the proposed mechanisms, starting on p. 14 (l. 504-527).

12) Lines 399-403: The role of founder cell stochasticity, including which species a cell encounters in a microhabitat and the initial number of cells of each species, could also account for the observed high diversity due to spatial fragmentation. This point warrants further exploration to differentiate its impact from that of phenotypic variation.

Reply and action: see reply under point 11.

13). Line 405: I am not sure I understand what is meant in the parenthesis:” dynamic variation in spatial fragmentation”

Reply: We wanted to suggest that not only the fragmentation alone, but also the variation over time of the fragmented structure (e.g. remixing of the soil matrix, wetting-drying cycles redistributing the micro-pockets of water and reshaping surface wetness) could be also relevant to the distribution of local interactions and emergent meta-communities composition.

Action: We clarified what we meant in the revised discussion in lines 549-563.

Figures:

14) Fig. 2A: *P. veronii* is still growing in monoculture at 24h. This suggests that the carrying capacity has not yet been reached. Clarifying whether this aspect influences the interpretation of competitive dynamics and outcomes would be beneficial

Reply and action: The culture fluorescence increases even after cells are in stationary phase because of continued expression from the promoters (l. 624) driving the gene for the fluorescent protein. This can be seen from a comparison of culture turbidity versus fluorescence measurements of the same well (Supplementary figure 3, mentioned in l. 163). For this reason, the fluorescence should not be used to infer yields in liquid culture (and we use flow cytometry cell counts). However, to distinguish both strains in co-culture and determine differences in their growth rates, we need fluorescence measurements.

15). Fig 3C: the symbols in the right are not immediately clear (mono-mixed).

Reply and action: This referred to the testing pairs (...). We explained this in the legend.

16) Fig. S8: I am not 100% convinced that founder cell variability does not have an impact on outcome.

*Reply: Yes, indeed. We meant to show that founder cell variability has an impact. That is why we regrouped the paired growth trajectories for droplets that started with the exact same number of *P. putida* and *P. veronii* founder cells (denoted by the two numbers on top of each facet plot), to highlight that variable growth kinetic could occur between droplets, despite having similar founder cell numbers. For example, the top right panel shows the growth trajectories of two droplets that started with 1 *P. putida* cells and 4 *P. veronii* cells.*

Action: We modified the title of Figure S8 to “Paired growth trajectories and founder cells number (PPU:PVE) ” to clarify the meaning of the numbers on top of each facet.

Minor/typos:

L. 105: missing parenthesis after “design from Duarte et al. 42”

Reply and action: Thank you. We corrected this.

L.191: “*veronii*, but again indifferent between mono- and co-culture conditions (Fig. 4C)”; I think it should refer to (Fig. 3C)?

Reply and action: Thank you very much for spotting this. We corrected the text.

Waiving anonymity

Nadav Kashtan

REVIEWERS' COMMENTS

Reviewer #1 (Remarks to the Author):

I thank the authors for their effort in preparing a new version of their manuscript including new experimental and numerical results, as well as for explaining their rationale in the rebuttal letter.

My main concern was the connection between the size of inoculi in their initial version of the manuscript (2-3 cells per droplet) and the inoculi reported in literature (as I perceived 100-1000 cells per habitat). The authors have now performed experiments and simulations to show that the effect of stochastic competition inversion in fact does diminish on the order of 10 initial cells. I have some doubts regarding some of the inoculum size estimates in various systems (see minor issues below), but importantly I do agree with the two statements that A) there is a range of inoculi in natural environments, and very small inoculi fall within this range (namely on leaf surfaces) and B) there is very little precise data on inoculum size, and therefore more research in this area is desirable. Therefore, I believe the authors have sufficiently addressed my main concern.

This is a fine piece of work. The quantitative experimental approach in combination with the mathematical model elevates the standards in microbial ecology. The manuscript is of general interest to people working on a range of systems and settings in microbial communities, as well as scientists working on stochastic phenomena in biology, and likely appeals to anyone interested in microbial interactions. I therefore recommend it for publication and the remaining points below are optional and can be considered at the discretion of the authors.

A few concrete suggestions to further improve the article, mostly textual and visual:

1. As I suggested in my initial report, the paper benefits from streamlining. The suggestion I made about removing the tailocin part in my initial report was only meant as an example. There is absolutely no doubt the authors spent a lot of time in preparing this extensive work, and this has resulted in a solid study. However, I believe the amount of work spent on certain items is not the most relevant metric in determining the devoted attention to each

study aspect. Although the current length and number of figures may technically comply with the journals requirements, the paper will be more read and better cited if the manuscript focusses on the results key to the main message of the paper (which I take to be the title). As a concrete example, the choice of the authors to append their additional work on different inoculum sizes in an additional figure has in my view only increased the need for streamlining (there must be a way for these to be presented together).

2. To contrast the microfluidic experiments better with the bulk experiments, it would be also good if the results are also presented in a symmetric way. I give one example: in a number of cases different metrics for the different types of experiments are used (like total productivity for the droplets and growth rate for the bulk experiments). These are obviously related, but it alleviates thinking in the readers' head if the exact same metric is reported, even if this requires more analysis steps. For example, what about extracting a total productivity for the bulk experiments and reporting this then side by side with the droplet experiments? Perhaps my example is wrong or I am missing something obvious here, but I hope the authors can see the point.

Minor issues

1. Is the inoculum really thousands or millions of cells in the case of the colonization of the gastrointestinal tract (LL 468)? I agree many (more like billions) of cells are entering the gut on a daily basis, but the point the authors make elsewhere in the study is that this bulk number is not the most relevant metric, it is the fragmentation of it. For example the mentioned crypts shielded by flow or the encapsulation by mucus. The typical size of these fragmented habitats sets the inoculum size, not the bulk intake. This can be connected better.

2. It has not escaped this reviewers' attention that the authors have removed the estimated inoculum size of marine particles. I can only guess the reason as they do not comment on this, but I think some kind of quantitative estimate would be useful for readers, even if preliminary. The inoculum size will for sure will depend on the marine particle size, and I suppose for sufficiently small particles a inoculum size <10 cells is imaginable.

3. Is "suspended-fragmented" really the best contrasting pair designator? What about "Fragmented-Continuous" or "suspended" can be replaced with "homogeneous" or "uniform". The authors have probably given this a lot more thought (and it is well defined in their manuscript), but I just feel that suspended is slightly off the mark and may confuse readers.

Reviewer #2 (Remarks to the Author):

I appreciate the substantial changes that improved an already great manuscript.

The authors have addressed the comments of myself and other reviewers with great care and I recommend publishing the manuscript in its current form. (A minute typo in the SI: Once spelled "Hutner" in the media recipe).

I appreciate the authors' additional work and willingness to perform further experiments.

Reviewer #2 (Remarks on code availability):

The authors provide all data and code from raw data, aggregation of the results, to visualization of the data.

Reviewer #3 (Remarks to the Author):

Overall, I find the revised version of the manuscript to be significantly improved. The authors have done an excellent job addressing the reviewers' comments and enhancing the manuscript.

The additional simulations and experiments, which include a wider range of droplet volumes and founder cell population sizes, are a valuable addition. They effectively

highlight the conditions under which phenotypic variation is expected to act or take place and demonstrate how and at what range of initial population sizes this effect diminishes.

The revised Discussion section is also much improved. It successfully emphasizes the main conclusions and their ecological relevance.

This is minor, but I still believe that presenting the competition dynamics in Supp Fig. 4 in a single graph, rather than in two separate graphs (i.e. for T0 and t24 separately), would be more informative. Specifically, for a pair of interacting species, A and B, such a graph displays the initial fraction of species A on the X-axis and the fraction of species A at the experiment's conclusion on the Y-axis. This approach offers a clearer understanding of the changes in the relative abundances of the competing species.

I have no further suggestions or comments and look forward to seeing this work published in Nature Communications.

REPLY TO THE REVIEWERS

Remaining reviewer comments:
Reviewer #1:

I thank the authors for their effort in preparing a new version of their manuscript including new experimental and numerical results, as well as for explaining their rationale in the rebuttal letter.

My main concern was the connection between the size of inoculi in their initial version of the manuscript (2-3 cells per droplet) and the inoculi reported in literature (as I perceived 100-1000 cells per habitat). The authors have now performed experiments and simulations to show that the effect of stochastic competition inversion in fact does diminish on the order of 10 initial cells. I have some doubts regarding some of the inoculum size estimates in various systems (see minor issues below), but importantly I do agree with the two statements that A) there is a range of inoculi in natural environments, and very small inoculi fall within this range (namely on leaf surfaces) and B) there is very little precise data on inoculum size, and therefore more research in this area is desirable. Therefore, I believe the authors have sufficiently addressed my main concern.

Reply: We thank the reviewer for the appreciation of the work.

This is a fine piece of work. The quantitative experimental approach in combination with the mathematical model elevates the standards in microbial ecology. The manuscript is of general interest to people working on a range of systems and settings in microbial communities, as well as scientists working on stochastic phenomena in biology, and likely appeals to anyone interested in microbial interactions. I therefore recommend it for publication and the remaining points below are optional and can be considered at the discretion of the authors.

A few concrete suggestions to further improve the article, mostly textual and visual:

1. As I suggested in my initial report, the paper benefits from streamlining. The suggestion I made about removing the tailocin part in my initial report was only meant as an example. There is absolutely no doubt the authors spent a lot of time in preparing this extensive work, and this has resulted in a solid study. However, I believe the amount of work spent on certain items is not the most relevant metric in determining the devoted attention to each study aspect. Although the current length and number of figures may technically comply with the journals requirements, the paper will be more read and better cited if the manuscript focusses on the results key to the main message of the paper (which I take to be the title). As a concrete example, the choice of the authors to append their additional work on different inoculum sizes in an additional figure has in my view only increased the need for streamlining (there must be a way for these to be presented together).

Reply: We thank the reviewer again for taking a second look at our work and appreciate their input and consent.

1) We have tried our best to streamline in the revision, but we believe it is important to keep the four interaction scenarios in the main text, since they are important for the developed argumentation. Since this aspect is at our discretion, we prefer to go ahead as is, and not do additional large edits to remove text parts to supplementary methods, text or other.

2. To contrast the microfluidic experiments better with the bulk experiments, it would be also good if the results are also presented in a symmetric way. I give one example: in a number of cases different metrics for the different types of experiments are used (like total productivity for the droplets and growth rate for the bulk experiments). These are obviously related, but it alleviates thinking in the readers' head if the exact same metric is reported, even if this requires more analysis steps. For example, what about extracting a total productivity for the bulk experiments and reporting this then side by side with the droplet experiments? Perhaps my example is wrong or I am missing something obvious here, but I hope the authors can see the point.

Reply: We appreciate this comment, but this is what we essentially do everywhere (except that growth rates cannot in all cases be determined in droplets since this would require time-lapse data). Examples, Fig. 2c and d (productivities in bulk and droplets), Fig. 3c (productivities in bulk and droplets). No further modifications needed.

Minor issues

1. Is the inoculum really thousands or millions of cells in the case of the colonization of the gastrointestinal tract (LL 468)? I agree many (more like billions) of cells are entering the gut on a daily basis, but the point the authors make elsewhere in the study is that this bulk number is not the most relevant metric, it is the fragmentation of it. For example the mentioned crypts shielded by flow or the encapsulation by mucus. The typical size of these fragmented habitats sets the inoculum size, not the bulk intake. This can be connected better.

Reply: We specified here that we mean 'food particles'.

2. It has not escaped this reviewers' attention that the authors have removed the estimated inoculum size of marine particles. I can only guess the reason as they do not comment on this, but I think some kind of quantitative estimate would be useful for readers, even if preliminary. The inoculum size will for sure will depend on the marine particle size, and I suppose for sufficiently small particles a inoculum size <10 cells is imaginable.

Reply: We removed this because among contacted experts there was no consensus as to what the founder cell populations on sinking marine snow particles is. There could be interparticle dispersal of bacteria, but also bacteria that have colonized e.g., individual protozoan or algae from the beginning, which then upon death becomes a sinking particle.

3. Is “suspended-fragmented” really the best contrasting pair designator? What about “Fragmented-Continuous” or “suspended” can be replaced with “homogeneous” or “uniform”. The authors have probably given this a lot more thought (and it is well defined in their manuscript), but I just feel that suspended is slightly off the mark and may confuse readers.

Reply: We thank the reviewer for this good suggestion. We added or explained 'uniform' to liquid suspended conditions throughout the text and figure.

Reviewer #2:

I appreciate the substantial changes that improved an already great manuscript. The authors have addressed the comments of myself and other reviewers with great care and I recommend publishing the manuscript in its current form. (A minute typo in the SI: Once spelled "Hutner" in the media recipe). I appreciate the authors' additional work and willingness to perform further experiments.

The authors provide all data and code from raw data, aggregation of the results, to visualization of the data.

Reply: We thank Reviewer 2 for the approval of our work.

Reviewer #3:

Overall, I find the revised version of the manuscript to be significantly improved. The authors have done an excellent job addressing the reviewers' comments and enhancing the manuscript.

The additional simulations and experiments, which include a wider range of droplet volumes and founder cell population sizes, are a valuable addition. They effectively highlight the conditions under which phenotypic variation is expected to act or take place and demonstrate how and at what range of initial population sizes this effect diminishes.

The revised Discussion section is also much improved. It successfully emphasizes the main conclusions and their ecological relevance.

Reply: We thank Reviewer 2 for the approval of our work.

This is minor, but I still believe that presenting the competition dynamics in Supp Fig. 4 in a single graph, rather than in two separate graphs (i.e. for T0 and t24 separately), would be more informative. Specifically, for a pair of interacting species, A and B, such a graph displays the initial fraction of species A on the X-axis and the fraction of species A at the experiment's conclusion on the Y-axis. This approach offers a clearer understanding of the changes in the relative abundances of the competing species.

Reply: We thank the reviewer for the suggestion. This is what we have essentially done in Fig. 6f. What we represent in Supp. Fig. 4 is really more the overall paired growth difference between T0 and T48. The reason why we cannot plot this as suggested is because these data are not paired – the same droplet is not followed between T0 and T24 (as in Fig 6f). Therefore, we would only be able to plot the changes in the ratio means across the replicates, and this will then mask the variation that we actually like to emphasize.

I have no further suggestions or comments and look forward to seeing this work published in Nature Communications.